# GENIU: A Restricted Data Access Unlearning for Imbalanced Data

## Abstract

With the increasing emphasis on data privacy, the significance of machine unlearning has grown substantially. Class unlearning, which involves enabling a trained model to forget data belonging to a specific class learned before, is important as classification tasks account for the majority of today's machine learning as a service (MLaaS). Retraining the model on the original data, excluding the data to be forgotten (also known as forgetting data), is a common approach to class unlearning. However, the availability of original data during the unlearning phase is not always guaranteed, leading to the exploration of class unlearning with restricted data access, which has attracted considerable attention. While current unlearning methods with restricted data access usually generate proxy sample via the trained neural network classifier, they typically focus on training and forgetting balanced data. However, the imbalanced original data can cause trouble for these proxies and unlearning, particularly when the forgetting data consists predominantly of the majority class. To address this issue, we propose the GENerative Imbalanced Unlearning (GENIU) framework. GENIU utilizes a Variational Autoencoder (VAE) to concurrently train a proxy generator alongside the original model. These generated proxies accurately represent each class and are leveraged in the unlearning phase, eliminating the reliance on the original training data. To further mitigate the performance degradation resulting from forgetting the majority class, we introduce an "in-batch tuning" strategy which works with the generated proxies. GENIU is the first practical framework for class unlearning in imbalanced data settings and restricted data access, ensuring the preservation of essential information for future unlearning. Experimental results confirm the superiority of GENIU over existing methods, establishing its effectiveness in empirical scenarios.

## 1    Introduction

Given the rising concerns on data privacy, and legal protections (European Parliament & Council of the European Union; BUKATY, 2019) the practice of machine unlearning (Nguyen et al., 2020; Brophy & Lowd, 2021; Sekhari et al., 2021), which allows a model to forget specific data, has become increasingly important. In specific, class unlearning has been considered significant to many real-world applications and can effectively addresses many privacy and usability needs, as classification services play an important role (Li et al., 2019; Guzella & Caminhas, 2009; Lu & Weng, 2007) in machine learning as a service (MLaaS) (Ribeiro et al., 2015). For example, in facial recognition, each individual's face is considered as a distinct class. Thus, when a model forgets a person's face, it essentially unlearns the class associated with that face (Masi et al., 2018). Similarly, in online shopping, products from a specific brand can be considered to all belong to an individual class – the brand. When a long-term customer of this brand loses interest, it is essential for the online shopping system to forget the customer's preference for this brand, i.e. unlearn the class quickly.

Generally, the class unlearning refers to a process of modifying or updating a well-trained model by forgetting or disregarding specific classes that it has learned previously. The data for the classes we want to forget is termed 'forgetting data', while the data for the classes we retain is called 'retaining data'. A straightforward unlearning method usually retrains a new model from scratch using the original data with the forgetting data excluded. Such *exact unlearning* (Bourtoule et al., 2021; Chen et al., 2022; Liu et al., 2021) is widely accepted but not efficient and requires the availability of full data which is challenging in real-world, i.e., SISA (Bourtoule et al., 2021), RecEraser (Chen et al.,

2022), and FedEraser (Liu et al., 2021). *Approximate unlearning* (Thudi et al., 2022; Graves et al., 2021) is usually more efficient as it focuses on updating parameters of the well-trained model to achieve class unlearning without the retraining of a new model, i.e., the Amnesiac (Graves et al., 2021) and Unrolling (Thudi et al., 2022). They all based on a strong assumption that the original data can be fully accessed during the unlearning phase. However, such assumption cannot hold in real-world applications due to considerations of storage efficiency and privacy. For example, in data sensitive applications, the original data will be deleted after the training for preserving data privacy. Also in some streaming service scenarios, data will not be saved for a long time due to the limited storage space. To combat the unavailability of the original data, generative based approximate unlearning methods such as zero-shot (Chundawat et al., 2023) and zero-glance (Tarun et al., 2021) unlearning have been proposed. Both of these approaches limit the retention of the original training data to some extent by employing a generative approach to create a limited set of proxies for each class. The generative method must be capable of producing proxies that faithfully capture the characteristics unique to each class. In the unlearning phase, such generative methods create class proxies to facilitate forgetting, and assume balanced data to ensure accurate class representation. However, in reality, there are a lot of scenarios when data are imbalanced (Spelmen & Porkodi, 2018; Rout et al., 2018). The presence of imbalanced data can significantly affect the performance of these generative methods by leading to biased representations and inadequate coverage of minority classes, resulting in suboptimal generation of proxies for those classes.

The challenge posed by imbalanced data becomes even more pronounced when examining existing approximate class unlearning methods such as Chundawat et al. (2023); Graves et al. (2021); Tarun et al. (2021). For the generative based methods (Chundawat et al., 2023; Tarun et al., 2021), as minority-class proxy samples might unintentionally carry characteristics of the majority class, the proxy samples may not accurately reflect the class characteristics sufficiently. This causes the model to use the unreliable proxies when unlearning, resulting in the inability to unlearning effectiveness. What is more, methods (Graves et al., 2021; Tarun et al., 2021) typically involve two steps: *impairment*, which erases the knowledge related to the forgetting data, and *repair*, which aims to restore performance on the retained data. If the majority class constitutes the forgetting data and is subjected to impairment, it results in the removal of a substantial portion of the model's task-specific knowledge, making it difficult to fully recover the performance on the remaining data.

To address the challenge of handling imbalanced data in class unlearning with limited data access, this study introduces a novel generative-based class unlearning approach. To tackle the issue of inaccurate proxy brought by imbalance, we present the innovative Generative Imbalanced Unlearning (GENIU) framework. Different with prior researches (Chundawat et al., 2023; Tarun et al., 2021), we leverage a generator structured with a Variational Autoencoder (VAE) (Kingma & Welling, 2014) which trained concurrently with the original model to produce reliable proxies for each class. Since the unlearning method cannot access data samples from original dataset, we employ carefully crafted noise samples, one for each class, as proxy generating prompts and will be stored for generating proxy with the trained generator in the unlearning phase. These noise samples are determined as designed class representations by the original model and rendered indistinguishable from human-generated data. This approach enhances privacy by thwarting attempts to recover features associated with the forgotten class. To further mitigate the adverse effects of unlearning the majority class on model performance, we introduce in-batch tuning. This technique simultaneously considers impairment and repair as a unified objective during the updating of the original model, contributing to a more effective and seamless unlearning process

Our contributions can be summarized as: 1) We are the first to explore the challenges presented by the application of data access restricted class unlearning methods within an imbalanced data setting. To the best of our knowledge, the proposed GENIU is also the first non-retrain-based unlearning framework for imbalanced data. 2) GENIU train the proxy generator and the original model at the same time, which ensures the generated proxy adequately represents its corresponding class by avoiding the minority class proxies unintentionally carrying the characteristics of the majority class. We also innovatively propose the *in-batch tuning* strategy during the unlearning phase to further mitigate the negative effect on the model performance as forgetting the majority class. 3) Through experimental results, we illustrate that existing unlearning methods, which restrict access to historical training data, struggle to perform well in an imbalanced data context. In contrast, GENIU shows superior performance over these baselines when tested on several widely used datasets, with high efficiency in both storage and time.

## 2 RELATED WORKS

**Machine unlearning.** Machine unlearning (Cao & Yang, 2015; Baumhauer et al., 2022; Nguyen et al., 2022a) is a new machine learning paradigm which allows data owners to completely delete their data from a machine learning model and enable their "right to be forgotten". Many existing unlearning works (Baumhauer et al., 2022; Brophy & Lowd, 2021; Cauwenberghs & Poggio, 2000; Chen et al., 2019; Mahadevan & Mathioudakis, 2021; Li et al., 2021) have found analytical optimization solutions by identifying the impact of data on model for traditional machine learning models, however, these unlearning methods are only suitable for machine learning methods with a convex problem nature. For deep neural networks in unlearning (Nguyen et al., 2022b), the non-convex nature of the problem and the stochasticity of the learning process have become the challenges which makes it hard to model the impact of data on the trained model and further eliminate such impact from model. A straightforward approach is to retrain a new model from scratch with a dataset that has no forgetting data. However, this retraining method is time-consuming, requires numerous data storage, and is infeasible when original training data is unavailable. To speed up the retraining process, SISA (Bourtoule et al., 2021) splits the complete dataset into several partitions and trains a model for each partition, thus it only needs to perform retraining on partitions that was containing unlearned data. Similar methods have been applied in recommender system (Chen et al., 2022) and federated learning (Liu et al., 2021) scenarios and this type of retrain-based method can be categorized as *exact unlearning*. Another type of method that requires no retraining of a new model from scratch is called *approximate unlearning*. The approximate unlearning can makes the parameters of the unlearned model closer to that of the retrained model by updating the original model for a few rounds. The Unrolling SGD (Thudi et al., 2022) and Amnesiac unlearning (Graves et al., 2021) record the changes of the parameter during the training of the data to be unlearned and recovers these changes during unlearning. However, all these methods require full access to the historical training data which cannot be satisfied in many real practices.

**Data restricted unlearning methods.** Most training data are often deleted or archived post-training due to storage costs and privacy concerns. Storing large amounts of data is expensive and poses security risks, especially with sensitive information. Data breaches or unauthorized access can lead to legal, ethical, and reputational consequences. Therefore, in a wider range of real practices, the unlearning method has no access to full or even partial of the historical training data. The zero-glance and zero-shot unlearning settings take such restrictions into account. The former can only access the retaining data in unlearning phase, while the latter is more strict and requires no access to any original data. The solutions corresponding to them, UNSIR (Tarun et al., 2021) and GKT (Chundawat et al., 2023) respectively, adopt the idea of generating proxies for the training data to provide a basis for unlearning. Detailedly, they use the well-trained classification model to generate proxies for inaccessible data, then use these proxies to represent actual data and perform unlearning. Therefore, these proxies trained through the knowledge of the well-trained classifiers are critical for unlearning. However, they both assume that the data used to train the original model is balanced. Due to data imbalance, the knowledge of the classifier can be biased, which in turn affects the generated proxies. Imbalanced data poses significant challenges to these generative-based methods as they may produce proxy samples for minority classes that inadvertently carry majority class traits, leading to unreliable unlearning.

**Learning and unlearning from imbalanced data.** An imbalanced dataset considers when there are some classes containing considerably more amount of samples (majority) than other classes (minorities). Learning from such an imbalanced dataset can make the predictions of minority classes inaccurate (Spelmen & Porkodi, 2018; Rout et al., 2018). An existing work (Koch & Soll) investigated the impact of imbalanced class setting on SISA (Bourtoule et al., 2021) unlearning method, when full original data is accessible during the unlearning phase they found that the imbalance in each data shard will lead corresponding retraining model unreliable. For example, in the case of imbalanced data, when the data is divided into various shards, some shards may be composed of the majority class or contain only a few samples of other classes. This will cause the model trained on this shard lacks or even has no data when retraining. This impact is more severe when access to training data is restricted, as less learning material is available for model retraining.

## 3 Preliminaries and Problem formalisation

In this section, we will first introduce preliminary notations and terms, i.e., class unlearning and imbalanced unlearning, and then formalise the problem of this work at the end of this section.

**Class unlearning.** Let $\mathcal{D} = \{(x_i, y_i)\}_{i=1}^n \in \mathcal{X} \times \mathcal{Y}$ be a dataset containing $n$ data samples that belong to $K$ classes. The $i$-th pair of the data sample and its associated label can be denoted as $(x_i, y_i)$, where $x_i \in \mathcal{X} \subseteq \mathbb{R}^d$ and $y_i \in \mathcal{Y} = \{1, \ldots, K\}$. We denote $\mathcal{D}^k = \{(x_i, y_i)|y_i = k\}$ as a subset of $\mathcal{D}$ that contains samples of the $k$-th class. When a class unlearning request is issued, it requires the classifier to forget knowledge on the forgetting class $\mathcal{Y}_f$ and maintain knowledge learned on the retain class $\mathcal{Y}_r$, where $\mathcal{Y}_f, \mathcal{Y}_r \subset \mathcal{Y}, \mathcal{Y}_f \cap \mathcal{Y}_r = \varnothing$ and $\mathcal{Y}_r \cup \mathcal{Y}_f = \mathcal{Y}$. Then, we can further denote their corresponding dataset $\mathcal{D}_f = \{(x_i, y_i|y_i \in \mathcal{Y}_f)\}$ and $\mathcal{D}_r = \{(x_i, y_i|y_i \in \mathcal{Y}_r)\}$, where $\mathcal{D}_f \cup \mathcal{D}_r = \mathcal{D}$ and $\mathcal{D}_f \cap \mathcal{D}_r = \varnothing$.

A deep learning neural network $f(x, \theta)$, which is parameterized by $\theta$, can output a vector $\boldsymbol{p} \in [0, 1]^K$, where the $j$th element of $\boldsymbol{p}$ represents the posterior probability of the $j$th label given $x$, i.e., $\boldsymbol{p}_j$ is interpreted as $P(y = j|x)$. In the context of unlearning, an original model $f(\cdot, \theta_{or})$ is trained with $\mathcal{D}$. A retrained model $f(\cdot, \theta_{re})$ is trained with $\mathcal{D}_r$. An unlearning method $\mathcal{U}$ is expected to make $f(\cdot, \theta_{or})$ forget the knowledge about $\mathcal{D}_f$ and output an unlearned model $f(\cdot, \theta_{un})$ which has the similar performance as a retrained model, i.e., $f(\cdot, \theta_{un}) \approx f(\cdot, \theta_{re})$. In retrain-based methods (Bourtoule et al., 2021; Chen et al., 2022; Liu et al., 2021), the unlearned model $f(\cdot, \theta_{un})$ is directly retrained with $\mathcal{D}_r$. However, as discussed above, they are computationally cost and infeasible when original training data is unavailable as retraining requires access to numerous training data to train a new model from scratch. Non-retrain methods (Thudi et al., 2022; Tarun et al., 2021; Chundawat et al., 2023), although more efficient, still assume that original data can be accessed when performing unlearning, i.e.,

$$f(\cdot, \theta_{un}) = \mathcal{U}(\mathcal{D}, f(\cdot, \theta_{or})). \tag{1}$$

**Imbalanced unlearning.** In the imbalanced unlearning setting, we assume the complete dataset $\mathcal{D}$ is imbalanced and contains a set of majority class, i.e., $\mathcal{Y}_m$. Then, we have $\mathcal{D}_m = \{(x_i, y_i)|y_i \in \mathcal{Y}_m\}$ that contains data of a majority class. We also have $\mathcal{D}_l = \{(x_i, y_i)|y_i \notin \mathcal{Y}_m\}$ that contains data of a class other than majority class. To facilitate the control of the variables, without special instructions, we assume all minority class have similar number of data and the number is far less than that of the majority class data. Then we have

$$|\mathcal{D}^{k_1}| \gg |\mathcal{D}^{k_2}| \ \forall k_1 \in \mathcal{Y}_m, \forall k_2 \notin \mathcal{Y}_m \ and \ |\mathcal{D}^{k_3}| \approx |\mathcal{D}^{k_4}| \ \ k_3 \neq k_4, k_3 \notin \mathcal{Y}_m, k_4 \notin \mathcal{Y}_m \tag{2}$$

The imbalance rate can be denoted as $r = |\mathcal{D}^{k_1}|/|\mathcal{D}^{k_2}|$, where $k_1 \in \mathcal{Y}_m, k_2 \notin \mathcal{Y}_m$. In this work, we assume that $\mathcal{D}_f$ contains one or more majority classes, that is $\mathcal{D}_m \subseteq \mathcal{D}_f$, which also means the unlearning request asks the model to forget the majority class(es).

**Target problem: class unlearning with restricted data access and imbalanced data setting.** Full access to $\mathcal{D}$ in the Eq 1 cannot be satisfied in many practical cases. Therefore, we follow the generative-based unlearning pipeline (Chundawat et al., 2023), which does not require the original training data and is applicable to a wider range of scenarios, using a set of generated proxy data $\mathcal{D}_p$ to provide approximate information about data features and make unlearning feasible.

We need to design an unlearning method $\mathcal{U}$ that, upon receiving an unlearning request which requires the forgetting of a majority class, i.e., the $k$-th class, is able to take the original model $f(\cdot, \theta_{or})$ as input and output an unlearned model $f(\cdot, \theta_{un})$ without using any data in $\mathcal{D}$, such that $f(\cdot, \theta_{un})$ is able to perform similarly to a model $f(\cdot, \theta_{re})$ retrained on data without the $k$-th class, i.e., the $\mathcal{D}_r$.

$$f(\cdot, \theta_{un}) = \mathcal{U}(\mathcal{D}_p, f(\cdot, \theta_{or})). \tag{3}$$

It is noteworthy that, unlike generative based unlearning, we aim at the situation where the $f(\cdot, \theta_{or})$ is learned from an imbalanced data distribution. It can be inferred from the Eq 3 that the proxy set $\mathcal{D}_p$ is critical for unlearning, and existing generative methods cannot generate $\mathcal{D}_p$ well enough in the situation of imbalanced data.

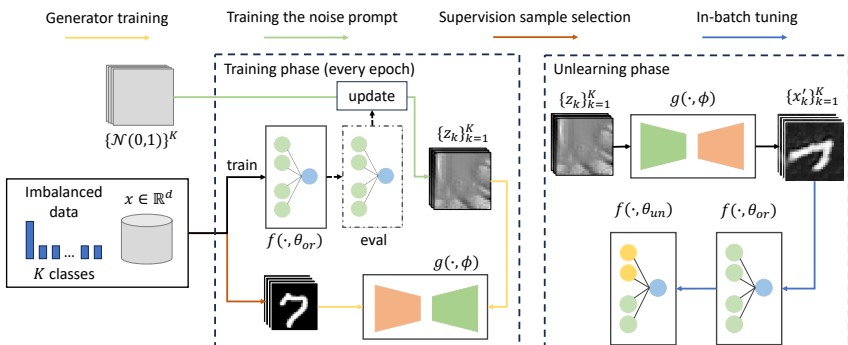

Figure 1: The overall view of GENIU. When the classifier $f(\cdot, \theta_{or})$ is training, noise prompt $z$'s are trained in the bypass. The proxy generator $g(\cdot, \phi)$ is also trained in this phase. In the unlearning phase, only the $z$'s and $g(\cdot, \phi)$ will be used to generate proxies $x'$ for unlearning using.

## 4  OUR METHOD

We show an overall view of GENIU in Figure 1. There are two main phases in GENIU i.e. the training phase and the unlearning phase. In the context of imbalanced data and no access to actual data samples, if the generator were trained by the well-trained $f(\cdot, \theta_{or})$ after the training phase, as existing works have done, the generated proxies cannot accurately represent the characteristic of its designed class, because most of the knowledge of $f(\cdot, \theta_{or})$ comes from the majority class and the generator learns some biased knowledge. Therefore, we need to record the correct feature when actual samples appear. We train and store the noise samples $\{z_i\}_{i=1}^{K}$ (one for each class) and a generator $g(\cdot, \phi)$ in the training phase to preserve valuable information about the features of the samples for proxy generating. In the unlearning phase, both $z$'s and $g(\cdot, \phi)$ will work together to generate reliable proxy samples, then a proposed in-batch tuning method will leverage these proxies to update the $f(\cdot, \theta_{or})$. This is a softer update method, other than the existing impair-repair update, that can eliminate the performance deduction on other knowledge when the model forgets most of the knowledge under the imbalanced unlearning problem. In the following subsections, we are going to detail these shown components one by one. Then, we provide the algorithms for both training and unlearning phase of the proposed GENIU.

### 4.1  PROXY GENERATOR

Under conditions of no access to original data, we need to generate proxies for original data to provide the information for unlearning. Considering the imbalanced data, existing proxy generating methods, which directly use the $f(\cdot, \theta_{or})$ as a guider and update a random noise sample with minimum error target, cannot get the proxies that can correctly express the characteristics of designed classes. Variational Autoencoders (VAE) Kingma & Welling (2014) is an impressive technology, in which the decoder can reconstruct a sample by giving a latent code and making the reconstructed sample $x'$ (also named as proxy in this work) look like a data sample in the training set. However, the belonged class of $x'$ depends on the given latent code. To generate data belonging to a particular class, the latent code needs to be specified. That is if we want to get a proxy $x'_i$, where $\{(x'_i, y'_i)|y'_i = k\}$, an ideal way is taking a real sample $x_i$ whose associated label $y_i = k$ as input of the generator's encoder and naturally get the appropriate code for the decoder. But it is infeasible when original data is unavailable. Therefore, we introduce a VAE structure as the proxy generator $g(\cdot, \phi)$ (Figure 2) and feed a carefully designed noise $z$ as a prompt for proxies' generating. The generating processing can be formalized as $x' = g(z, \phi)$, where $z$ is the carefully designed noise which can be determined as a designed class by $f(\cdot, \theta_{or})$ and will be detailed in Section 4.2.

It is difficult to train the $g(\cdot, \phi)$ in the unlearning phase, because the knowledge of the $g(\cdot, \phi)$ cannot be accurately obtained in the unlearning phase as there are no samples available that can accurately describe the class characteristics. Thus, we intend to train the generator in the training phase alongside the training of $f(\cdot, \theta)$. Detailedly, given a set of noise $\mathcal{D}_z = \{(z_k, y_k)|y_k = k\}_{k=1}^{K}$ which contains only $K$ pairs of noise and label, and a set of selected samples from training dataset

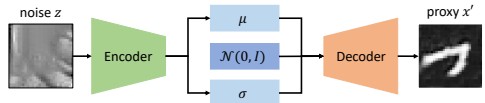

Figure 2: The proxy generator $g(\cdot, \phi)$ used in GENIU.

$\mathcal{D}_s = \{(x_k, y_k)|y_k = k\}_{k=1}^{K}$. The reconstruction loss $\mathcal{L}_{rec}$ can be defined as

$$\mathcal{L}_{rec} = \frac{1}{K} \sum_{k=1}^{K} \|g(z_k, \phi) - x_k\|. \tag{4}$$

To make the learned Gaussian distribution more accurate, a distribution loss $\mathcal{L}_{dis}$ can be defined as

$$\mathcal{L}_{dis} = \frac{1}{2K} \sum_{k=1}^{K} \sum_{j=1}^{l} (1 + \log((\sigma_j^k)^2) - (\sigma_j^k)^2 - (\mu_j^k)^2) \tag{5}$$

where the $\mu \in \mathbb{R}^l$ and $\sigma \in \mathbb{R}^l$ are learnable gaussian distribution parameters for modeling the laten code, and the $l$ is the dimension of the latent code. Finally, the overall objective of learning generator $g(\cdot, \phi)$ is

$$\min_{\phi} \mathcal{L}_{gen} = \mathcal{L}_{rec} - \lambda \mathcal{L}_{dis} \tag{6}$$

where $\lambda$ is a hyperparameter that used to trade-off the impact of $\mathcal{L}_{rec}$ and $\mathcal{L}_{dis}$. Optimizing Eq. 6 could give the generator. The details on how to select $x_k$'s will be introduced in Section 4.4.

## 4.2 TRAINING THE NOISE PROMPT.

To avoid using a historical data sample as a guide to reconstruct a proxy samples of a specific class in the unlearning phase, we intend to train a noise $z_k$ as the prior knowledge for constructing a proxy sample of the specific class $k$. Specifically, the trained noise $z_k$ should be correctly determined as the interesting class $k$ by the classifier $f(\cdot, \theta)$, that is $y_k = f(z_k, \theta)$ and $y_k = k$. To achieve this goal, we update a randomly initialized noise $z_{init}$ by minimizing the classification error, which satisfies

$$z_{init} \in \mathbb{R}^d \sim \mathcal{N}(0, 1) \in \mathbb{R}^d. \tag{7}$$

The optimization objective of noise $z_k$ is basically the original task objective. For the classification task, this objective should be

$$z_k = \min_{z} CrossEntropy(f(z, \theta), y_k), \quad y_k = k. \tag{8}$$

In this work, we use the Adam optimizer (Kingma & Ba, 2015) to update the randomly initialized noise $z_{init}$ according to the objective equation 8. It is worth noting that noise and classifier are updated independently, and the training of noise will not affect the training of the classifier.

## 4.3 IN-BATCH TUNING FOR UNLEARNING.

To further mitigate the performance degradation of the model by forgetting the majority class in the imbalance unlearning, we make the mini-batch of each unlearning step containing proxies of each class. It is noteworthy that, in the unlearning phase, we need only one mini-batch which includes $K$ proxies. A proxy $x'_k$ is generated by $g(\cdot, \phi)$ with a given trained noise $z_k$. Therefore, the dataset used for unlearning is

$$\mathcal{D}_u = \{(x'_k, y_k)|y_k = k\}_{k=1}^{K}, \quad where \ x'_k = g(z_k, \phi). \tag{9}$$

In the process of model tuning, we hope that the proxies that need to be unlearned can make the model change in the direction of increasing error, and the proxies that need to be retained can make the model continue to change on the direction of reducing error. In consideration of this, we design the following loss

$$\mathcal{L}_u = \sum_{(x'_k, y_k) \in \mathcal{D}_u, y_k \in \mathcal{Y}_r} \mathcal{L}(f(x'_k, \theta), y_k) + \sum_{(x'_k, y_k) \in \mathcal{D}_u, y_k \in \mathcal{Y}_f} \frac{1}{\mathcal{L}(f(x'_k, \theta), y_k)}, \tag{10}$$

where the used loss $\mathcal{L}(\cdot, \cdot)$ should be the same as the loss on which the original model is trained.

## 4.4 SUPERVISION SAMPLE SELECTION.

Since we use a tuning style method to perform unlearning only with generated proxies $x'$, if the $x'$ can be correctly classified by $f(\cdot, \theta)$ with high confidence, the tuning step would be small, since in this situation the $x'$ is away from the decision boundary and results in a small value of classification loss. Therefore, we prefer the selected supervision samples $x_k$ (Eq.4) near to the decision boundary. Specifically, we select an $x$ with maximum logit entropy for each class. The logits entropy $E(x)$ can be calculated as $E(x) = -\sum_{k=1}^{K} p_k \cdot \log(p_k)$, where the $p_k$ is the output probability of $x$ belonging to the $k$-th class. The higher the $E(x)$, the closer each probability in $p$ and also the higher the uncertainty of determining $x$. Therefore, to supervise the training of $g(\cdot, \phi)$, we need a set of supervision samples $\mathcal{D}_s$, whose items are selected as $x_k = \max_{x_i \in \mathcal{D}} E(x_i)$ and $y_i = k$.

## 4.5 GENIU ALGORITHM.

The proposed GENIU is divided into *training phase* and *unlearning phase*. During the training phase (Appendix. A, Algorithm. 1), the classifier $f(\cdot, \theta)$ will be trained normally. In each epoch of $f(\cdot, \theta)$ training, additional training on noise $z$'s is performed. If the trained noise $z$'s in an epoch can be correctly classified by $f(\cdot, \theta)$, these noises will be used together with the selected sampled $x$'s to train the generator $g(\cdot, \phi)$, otherwise the training of the generator will be skipped in this epoch.

In the unlearning phase (Appendix. A, Algorithm. 2), only the trained noise $z$'s and generator $g(\cdot, \phi)$ will be used. The generator will reconstruct $z$ into proxy $x'$, and then the in-batch tuning will use these proxies to adjust $f(\cdot, \theta_{or})$ and finally output the unlearned model $f(\cdot, \theta_{un})$.

## 5 EXPERIMENTS

**Datasets.** We evaluate the effectiveness of the proposed GENIU on four benchmark datasets, i.e., Digits-MNIST (LeCun et al., 1998), Fashion-MNIST (Xiao et al., 2017), Kuzushiji-MNIST (Clanuwat et al., 2018) and CIFAR-10 (Krizhevsky et al., 2009). Detailedly, all these three MNIST style dataset contains 60,000 samples in their training set and 10,000 samples in their test set. Each sample of these MNIST style dataset is a $28 \times 28$ grayscale image associated with a label from ten classes. In the Digits-MNIST, the classes are handwritten digits from 0 to 9. In the Fashion-MNIST, the classes are ten different fashion items (i.e. T-shirts, shoes). In the Kuzushiji-MNIST, the classes are ten different Hiragana characters. CIFAR-10 contains 50,000 training samples and 10,000 test samples each of which is an RGB image in the shape of $32 \times 32$ and associated with one of ten semantic classes. To make the imbalanced dataset, we set the imbalance rate $r = 0.1$ in this work. Specifically, we keep the number of samples of majority class the same as the raw dataset and select 10% samples for each of the minority classes.

**Baselines.** We conduct comparison experiments on two types of methods, one can access the original data that includes I-R (Graves et al., 2021) and Unrolling SGD (Thudi et al., 2022), the other cannot access the original data that includes GKT (Chundawat et al., 2023) and UNSIR (Tarun et al., 2021). In expectation, methods which can access training data should have better performance than methods cannot access training data. Specifically, 1) I-R (Graves et al., 2021), Amnesiac records the changes of the parameter during the training of the data to be unlearned and recovers these changes during unlearning. 2) Unrolling SGD (Thudi et al., 2022). In unlearning phase, it arranges forgetting data in the first batch and performs incremental training with both unlearned training data and retain training data. It records gradients when learning the first batch and adds recorded gradients on weights after the incremental training. 3) GKT (Chundawat et al., 2023), which is the SOTA zero-shot unlearning method. The GKT generates the error maximized noise to proxy $\mathcal{D}_f$ and generates error minimized noise to proxy $\mathcal{D}_r$. Then, it initializes a new network called the student and teaches the student with the original model. 4) UNSIR (Tarun et al., 2021), which is the SOTA zero-glance unlearning method. It generates the error maximized noises to proxy $\mathcal{D}_f$ and mixes these noises with a part of $\mathcal{D}_r$. Then, it performs impair-repair steps to tune the original model.

**Implementation details.** For all experiments, we use the AllCNN (Springenberg et al., 2015) as the base classification model as it has been widely used for image data and been used by baselines. Following the baselines' setting, the training batch size for all dataset is set as 256, and the learning

rate and weight decay are set as 0.01 and $10^{-4}$, respectively. We also follow the default setting in the VAE (Kingma & Welling, 2014) and set the learning rate for training of noise $z$ and generator $g(\cdot, \phi)$ as 0.02 and 0.005, respectively, and $\lambda = 2.5 \times 10^{-4}$ (Eq.6). Then, 1) for all **MNIST** style datasets, in the training phase, we train the AllCNN for 20 epochs, and train the initialized noise $z$ as well as the generator $g(\cdot, \phi)$ for 100 steps in each epoch. In the unlearning phase, we conduct in-batch tuning for 100 rounds. 2) For **CIFAR-10**, in the training phase, we train the AllCNN for 40 epochs, and train the initialized noise $z$ for 100 steps and the generator $g(\cdot, \phi)$ for 200 steps in each epoch. In the unlearning phase, we conduct in-batch tuning for 45 rounds. For all dataset, in the unlearning phase, we set the learning rate for tuning $f(\cdot, \theta_{or})$ as $4 \times 10^{-4}$. In the generator, we set a CNN structure with increasing channels for the encoder, i.e. [32, 64, 128, 256], and the decoder is a CNN structure symmetrical to the encoder. The dimension of the latent code is 128 for MNIST style dataset and 256 for CIFAR-10. All other parameters of baseline methods follow their default settings. All the experiments are conducted with NVIDIA RTX A5000 GPU and the reported results are the average of five trials of experiments using different seeds.

## 5.1 RESULTS AND ANALYSIS

Table 1: Unlearning performance. The direction of the arrow indicates the desired direction of value change. The up arrow means higher is better, the down arrow means lower is better.

| Dataset | Acc | Original Model | Retrain Model | GKT | UNSIR | GENIU (ours) | I-R | Unrolling |
|---|---|---|---|---|---|---|---|---|
| D-MNIST | $\mathcal{D}_r \uparrow$ | 0.9494 | 0.9405 | 0.4116 | 0.3502 | **0.9286** | 0.9766 | 0.8555 |
| | $\mathcal{D}_f \downarrow$ | 0.9913 | 0.0 | 0.0258 | 0.0001 | 0.0065 | 0.0 | 0.1466 |
| F-MNIST | $\mathcal{D}_r \uparrow$ | 0.8057 | 0.816 | 0.2595 | 0.3002 | **0.7711** | 0.8368 | 0.7571 |
| | $\mathcal{D}_f \downarrow$ | 0.9681 | 0.0 | 0.0 | 0.0016 | 0.0002 | 0.0 | 0.4015 |
| K-MNIST | $\mathcal{D}_r \uparrow$ | 0.8772 | 0.8641 | 0.3537 | 0.2346 | **0.7012** | 0.8788 | 0.8073 |
| | $\mathcal{D}_f \downarrow$ | 0.9764 | 0.0 | 0.0029 | 0.0 | 0.0004 | 0.0 | 0.0550 |
| CIFAR-10 | $\mathcal{D}_r \uparrow$ | 0.5952 | 0.6347 | 0.273 | 0.1778 | **0.4948** | 0.4838 | 0.3971 |
| | $\mathcal{D}_f \downarrow$ | 0.9452 | 0.0 | 0.0 | 0.0327 | 0.0103 | 0.0 | 0.0136 |

**Effectiveness.** We conduct unlearning experiments with each class as forgetting class (majority class) on each dataset and report the mean accuracy performance in Table 1. From the performance of the original model on $\mathcal{D}_r$ and $\mathcal{D}_f$, it can be seen that the imbalanced dataset will cause a corresponding imbalance in the performance of the original model. The model will perform significantly better in the majority class than in other classes. Among all methods with limited access to original data, the proposed method GENIU performs best. GKT and UNSIR, relying on the original model for proxy generation, their generated proxies are affected by this imbalance, impacting unlearning quality. I-R and Unrolling, with full historical data access, generally outperformed GENIU, but GENIU showed better results on CIFAR-10. Detailed results from Fashion-MNIST (Appendix B, Table 6) demonstrate unlearning performance when each class is the majority. Further tests with multiple classes as majority for deletion (0-th and 1-st classes) also confirm these findings, as reported in Appendix C (Table 7).

**Why existing generative based unlearning methods failed with imbalanced data?** To further prove that GENIU can obtain more reliable noise in the case of imbalance data, we try to observe the origin model's perception on noises generated by different methods. Intuitively, the noise generated by leveraging the well-trained $f(\cdot, \theta_{or})$ will have the characteristics of the majority class, since the knowledge from majority class dominates the model. Therefore, the origin model's perception of noise of other classes will be closer to that of the majority class. Specifically, the distribution of the model's logits output of minority classes will be closer to that of the majority class. To verify this, we sample some training examples of the majority class and feed them to $f(\cdot, \theta_{or})$ to obtain **reference perception** $p_{ref}$, which is basically the output logits. Then, we feed the noise of other classes generated by different unlearning methods to $f(\cdot, \theta_{or})$ to obtain **observation perception** $p_{obs}$. We then try to fit the distribution of the reference perception with the distribution of these observation perceptions, which is a common use of KL divergence $D_{kl}(p_{obs}||p_{ref})$, to observe the difference between the $p_{obs}$ and the $p_{ref}$ of different methods. According to the property of KL divergence, the greater the $D_{kl}(p_{obs}||p_{ref})$, the more significant the difference between the model's perception of generated noise and its perception of the majority class, that is, the better. Since UNISR only generates noise for the forgetting class and does not generate noise for other classes, here we only compare the GKT and GENIU methods. From Table 2, we can observe that when producing noise

for the four data sets, the $D_{kl}(p_{obs}||p_{ref})$ of the noise generated by GENIU is greater than that of GKT. This shows that the origin model's perception of the noise generated by GKT is closer to the majority class, and it carries more characteristics of the majority class than the noise generated by GENIU. We also reconstruct more specific proxies for GKT by using its generated noises and the trained VAE of GENIU. Since the noise generated by GKT carries more features of the majority class, these reconstructed proxies will make these features more specific. As can be seen from the Table 3, it is more difficult for GKT to use such reconstructed proxies to eliminate the knowledge of the majority class. Some visualized samples are provided in Appendix D.

Table 2: Comparing origin model's perception of noise generated by different methods in $D_{kl}(p_{obs}||p_{ref})$.

| Noise Generator | D-MNIST | F-MNIST | K-MNIST | CIFAR-10 |
|---|---|---|---|---|
| GKT | 11.7565 | 11.4835 | 12.6639 | 12.2472 |
| GENIU | **12.2526** | **11.8418** | **13.2708** | **12.9941** |

Table 3: Reconstruct proxy with noise generated by existing method.

| Method | Acc | D-MNIST | F-MNIST | K-MNIST | CIFAR-10 |
|---|---|---|---|---|---|
| GKT_vae | $\mathcal{D}_r \uparrow$ | 0.6115 | 0.4854 | 0.269 | 0.1429 |
|  | $\mathcal{D}_u \downarrow$ | 0.7567 | 0.8418 | 0.514 | 0.687 |
| GENIU | $\mathcal{D}_r \uparrow$ | **0.9286** | **0.7711** | **0.7012** | **0.4948** |
|  | $\mathcal{D}_u \downarrow$ | **0.0065** | **0.0002** | **0.0004** | **0.0103** |

Table 4: Time cost in unlearning phase.

| Dataset | Time cost | GKT | UNSIR | GENIU | I-R | Unrolling |
|---|---|---|---|---|---|---|
| D-MNIST | ms | 39086 | 1804 | **326** | 17005 | 483 |
| F-MNIST | ms | 39702 | 1854 | **327** | 16848 | 608 |
| K-MNIST | ms | 37312 | 1758 | **330** | 16254 | 411 |
| CIFAR-10 | ms | 33633 | 2515 | **159** | 16601 | 195 |

Table 5: Main contribution ablation.

| Proxy | Tuning | $Acc_u$ | $Acc_r$ |
|---|---|---|---|
| Post | Impair-Repair | 0.123 | 0.27 |
| GENIU | Impair-Repair | 0.048 | 0.758 |
| Post | GENIU | 0.018 | 0.416 |
| GENIU | GENIU | **0.0** | **0.771** |

**Unlearning efficiency.** We compare the time consumption among various unlearning methods. Experiments were conducted under identical conditions, measuring the time in milliseconds from inputting the original model $f(\cdot, \theta_{or})$ to outputting the unlearned model $f(\cdot, \theta_{un})$. Results in Table 4 show that GENIU is more time-efficient in the unlearning phase, as it doesn't require training a generation network and only uses a small number of proxies equal to the class count for adjustments. Regarding storage costs, retaining original data for a MNIST-like dataset needs 45MB, and CIFAR-10 needs 169MB. But storing a generator instead requires only 4.6MB for MNIST and 6.1MB for CIFAR-10.

**Ablation studies** We also conduct ablation studies to assess the impact of different technologies on GENIU. It starts by evaluating two main techniques, as shown in Table 5. Further investigations focus on the type of supervision sample selection and the number of in-batch tuning rounds, with findings and analyses detailed in Appendices G.2 and G.3.

The study examines how two technical components affect unlearning performance: 1) training a proxy generator alongside the original model, and 2) in-batch tuning during the unlearning phase. It compares the first with post-training generated proxies and the second with an impair-repair process, using identical learning rates and rounds. From the results which are reported in Table 5, when the proxy generated by the GENIU framework is applied, the impair-repair process will first forget the knowledge related to the majority class, however, this part of knowledge is most of the knowledge of the model about the classification task and makes the model hard to maintain the performance on retain classes in the subsequent repair stage. Additionally, when using post-training generated proxies, the imbalance in original training data causes these proxies to exhibit characteristics of the majority class, reducing the model's ability to distinguish between classes to be retained after forgetting the majority class.

# 6 CONCLUSION

In this work, we explore the challenges presented by the applications of restricting data access unlearning methods within an imbalanced data setting. The proposed framework, Generative Imbalanced Unlearning (GENIU) offers an effective solution to these challenges. GENIU requires neither training a new model from scratch nor access to any historical training data. The unique approach of training the proxy generator and the original model concurrently ensure the proxies accurately represent their corresponding classes. The in-batch tuning strategy that we introduce in the unlearning phase effectively mitigates the performance degradation as the model unlearns the majority class. The experimental results confirm GENIU's superior performance over existing methods, demonstrating its practicality and efficiency within the imbalanced data setting.

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

## A  GENIU ALGORITHM

The proposed GENIU is divided into *training phase* and *unlearning phase*. During the training phase (Algorithm. 1), the classifier $f(\cdot, \theta)$ will be trained normally. In each epoch of $f(\cdot, \theta)$ training, additional training on noise $z$'s is performed. If the trained noise $z$'s in an epoch can be correctly classified by $f(\cdot, \theta)$, these noises will be used together with the sampled $x$'s to train the generator $g(\cdot, \phi)$, otherwise the training of the generator will be skipped in this epoch.

---
**Algorithm 1** Training phase
---
**Require:**
    Dataset $\mathcal{D}$ and number of classes $K$;
    Initialized noise $\{z_i\}_{i=1}^K$ and rounds $Epoch_z$;
    Initialized classifier $f(\cdot, \theta)$ and epochs $Epoch_f$;
    Initialized generator $g(\cdot, \phi)$ and rounds $Epoch_g$;
1: **for** $e_n$ in $range(Epoch_f)$:
2:     Train $f(\cdot, \theta)$;
3:     **for** $e_z$ in $range(Epoch_z)$:
4:         Train $\{z_i\}_{i=1}^K$ (section 4.2);
5:     **if** $\{z_i\}_{i=1}^K$ can be correctly classified by $f(\cdot, \theta)$:
6:         Select supervision $\{x_i\}_{i=1}^K$ (section 4.4);
7:         **for** $e_g$ in $range(Epoch_g)$:
8:             Train $g(\cdot, \phi)$ with $\{z_i\}_{i=1}^K$ and $\{x_i\}_{i=1}^K$;
9: **return** $\{z_i\}_{i=1}^K$, $g(\cdot, \phi)$ and $f(\cdot, \theta_{or})$;
---

In the unlearning phase (Algorithm. 2), only the trained noise $z$'s and generator $g(\cdot, \phi)$ will be used. The generator will reconstruct $z$ into proxy $x'$, and then in-batch tuning will use these proxies to adjust $f(\cdot, \theta_{or})$ and finally output the unlearned model $f(\cdot, \theta_{un})$.

---
**Algorithm 2** Unlearning phase
---
**Require:**
    Trained noise $\{z_i\}_{i=1}^K$;
    Trained generator $g(\cdot, \phi)$;
    Trained classifier $f(\cdot, \theta_{or})$;
    Unlearning rounds $Epoch_u$;
1: $x' = g(z, \phi)$;
2: **for** $e_u$ in $range(Epoch_u)$:
3:     Do in-batch tuning with $x'$ (section 4.3);
4: **return** $f(\cdot, \theta_{un})$;
---

## B  DETAILED ACCURACY PERFORMANCE

A more detailed example result, which is from the Fashion-MNIST dataset and shows the unlearning performance when each class is set as forgetting class (majority class), is listed in Table 6. From the results, we can see that the proposed GENIU outperforms both GKT and UNSIR. In addition, we can also observe that there are differences in the unlearning performance when different classes are set as unlearned classes. This may be related to the relationship between the features of the forgetting class and the features of the retain class. However, the influence of the relationship between the features of the forgetting class and the features of the retain class on the unlearning performance is still unclear and remains to be explored.

## C  MULTICLASS UNLEARNING WHEN IMBALANCED DATA

We also test all methods when there are multiple classes of data that are majority class and needed to be deleted. We set the 0-th and 1-st classes of each dataset are majority classes that are needed

Table 6: Detailed accuracy performance from Fashion-MNIST.

| Class | Acc | Original Model | Retrain Model | GKT | UNSIR | GENIU (ours) | I-R | Unrolling |
|---|---|---|---|---|---|---|---|---|
| T-shirt/top | $\mathcal{D}_r \uparrow$ | 0.796 | 0.8544 | 0.2602 | 0.2453 | **0.784** | 0.8209 | 0.8636 |
| | $\mathcal{D}_f \downarrow$ | 0.9388 | 0.0 | 0.0 | 0.0 | 0.0 | 0.0 | 0.4618 |
| Trouser | $\mathcal{D}_r \uparrow$ | 0.7469 | 0.8376 | 0.2771 | 0.2962 | **0.7427** | 0.8120 | 0.7751 |
| | $\mathcal{D}_f \downarrow$ | 0.9758 | 0.0 | 0.0 | 0.0112 | 0.0 | 0.0 | 0.8096 |
| Pullover | $\mathcal{D}_r \uparrow$ | 0.8111 | 0.8422 | 0.1651 | 0.2327 | **0.7427** | 0.8758 | 0.8764 |
| | $\mathcal{D}_f \downarrow$ | 0.9784 | 0.0 | 0.0 | 0.0 | 0.0 | 0.0 | 0.63 |
| Dress | $\mathcal{D}_r \uparrow$ | 0.8424 | 0.8447 | 0.2851 | 0.2938 | **0.7993** | 0.8285 | 0.8278 |
| | $\mathcal{D}_f \downarrow$ | 0.9734 | 0.0 | 0.0 | 0.0034 | 0.0 | 0.0 | 0.3574 |
| Coat | $\mathcal{D}_r \uparrow$ | 0.8238 | 0.8089 | 0.2416 | 0.4018 | **0.8287** | 0.8511 | 0.8687 |
| | $\mathcal{D}_f \downarrow$ | 0.9518 | 0.0 | 0.0 | 0.0 | 0.0004 | 0.0 | 0.26 |
| Sandal | $\mathcal{D}_r \uparrow$ | 0.8184 | 0.7416 | 0.2684 | 0.2004 | **0.6996** | 0.8244 | 0.8549 |
| | $\mathcal{D}_f \downarrow$ | 0.9948 | 0.0 | 0.0004 | 0.0 | 0.0 | 0.0 | 0.6552 |
| Shirt | $\mathcal{D}_r \uparrow$ | 0.7996 | 0.8878 | 0.1782 | 0.2671 | **0.7898** | 0.8911 | 0.8073 |
| | $\mathcal{D}_f \downarrow$ | 0.9068 | 0.0 | 0.0 | 0.0 | 0.0 | 0.0 | 0.016 |
| Sneaker | $\mathcal{D}_r \uparrow$ | 0.8218 | 0.8093 | 0.3742 | 0.3491 | **0.7529** | 0.8124 | 0.8504 |
| | $\mathcal{D}_f \downarrow$ | 0.9948 | 0.0 | 0.0 | 0.0012 | 0.0 | 0.0 | 0.003 |
| Bag | $\mathcal{D}_r \uparrow$ | 0.7724 | 0.7396 | 0.2758 | 0.3918 | **0.7749** | 0.8304 | 0.0139 |
| | $\mathcal{D}_f \downarrow$ | 0.987 | 0.0 | 0.0 | 0.0 | 0.0 | 0.0 | 0.557 |
| Ankle boot | $\mathcal{D}_r \uparrow$ | 0.8244 | 0.7942 | 0.2696 | 0.3236 | **0.7964** | 0.8218 | 0.8327 |
| | $\mathcal{D}_f \downarrow$ | 0.9798 | 0.0 | 0.0 | 0.0 | 0.0016 | 0.0 | 0.2648 |

Table 7: Multi-class unlearning results. The 0-th and 1-st class of each dataset are set as forgetting class (majority class).

| Dataset | Acc | Original Model | Retrain Model | GKT | UNSIR | GENIU (ours) | I-R | Unrolling |
|---|---|---|---|---|---|---|---|---|
| D-MNIST | $\mathcal{D}_r \uparrow$ | 0.9515 | 0.9755 | 0.5429 | 0.4682 | **0.9018** | 0.9468 | 0.4805 |
| | $\mathcal{D}_f \downarrow$ | 0.9967 | 0.0 | 0.5168 | 0.0 | 0.0002 | 0.0 | 0.0 |
| F-MNIST | $\mathcal{D}_r \uparrow$ | 0.7857 | 0.7975 | 0.1755 | 0.3303 | **0.7333** | 0.8119 | 0.7498 |
| | $\mathcal{D}_f \downarrow$ | 0.9807 | 0.0 | 0.4711 | 0.0919 | 0.052 | 0.0 | 0.1416 |
| K-MNIST | $\mathcal{D}_r \uparrow$ | 0.8422 | 0.8658 | 0.3993 | 0.3222 | **0.649** | 0.8119 | 0.6811 |
| | $\mathcal{D}_f \downarrow$ | 0.9722 | 0.0 | 0.787 | 0.0 | 0.0 | 0.0 | 0.0166 |
| CIFAR-10 | $\mathcal{D}_r \uparrow$ | 0.5795 | 0.6425 | 0.3485 | 0.0998 | **0.528** | 0.5010 | 0.4104 |
| | $\mathcal{D}_f \downarrow$ | 0.9548 | 0.0 | 0.7849 | 0.1997 | 0.045 | 0.0 | 0.0152 |

to be deleted and report the results in Table 7. From the results, we can observe that the proposed GENIU can still achieve the best performance among methods that have restricted data access. It is worth noting that when the GKT method performs multi-class unlearning under imbalanced data settings, it still has relatively high accuracy on forgetting data. This is because GKT generates proxies for both unlearning and retraining data, and filters the unlearning proxy through the designed knowledge gate, however, the retained proxies through the gate may still represent features that are related to the forgetting class.

## D   VISUALIZING NOISE AND RECONSTRUCTED PROXIES

In this subsection, we visualize the noise generated by different methods, as well as the samples reconstructed using these noises and VAE trained by GENIU. From Figure 3 we can see that when the 9-th class is set as the majority class, the noise generated by GKT for other classes cannot accurately reflect the characteristics of their respective classes, and the characteristics of the majority class can be seen from their reconstructed images. For example, in the D-MNIST dataset, the reconstructed samples of the number 1 show curved characteristics, and the reconstructed samples of the number 3 look more like the number 9. In the F-MNIST dataset, the reconstructed samples of some classes look like majority class (Ankle boot), the highlighted pixels of reconstructed samples of other classes are also concentrated in the lower half of the image. However, we can see from Figure 4 that the reconstructed samples via the noise generated by GENIU can visually reflect the characteristics of their corresponding classes.

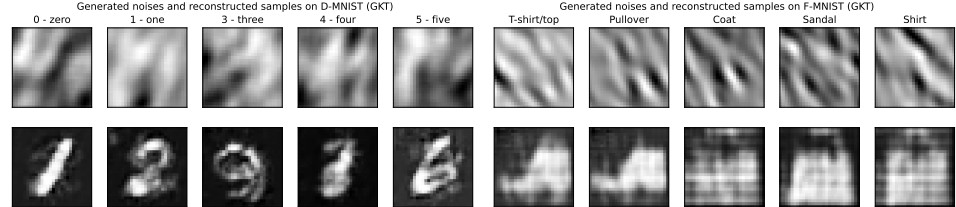

Figure 3: The figure shows the noises learned by GKT (upper) and corresponding VAE reconstructed samples (lower) of some classes in the D-MNIST and F-MNIST dataset. The 9-th class (digit number 9 for D-MNIST and Ankle boot for F-MNIST) is set as majority class for both dataset.

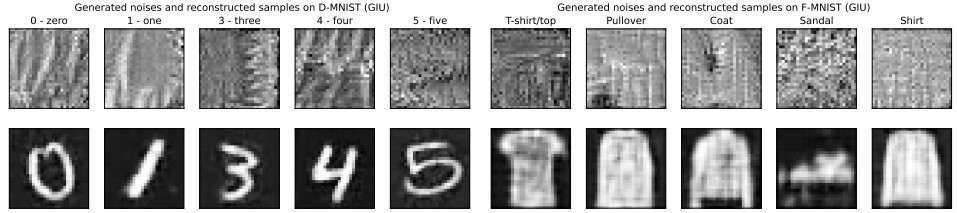

Figure 4: The figure shows the noises learned by GENIU (upper) and corresponding VAE reconstructed samples (lower) of some classes in the D-MNIST and F-MNIST dataset. The 9-th class (digit number 9 for D-MNIST and Ankle boot for F-MNIST) is set as majority class for both dataset.

# E    TIME COST OF AUXILIARY MODELS

As we place the training of the proxy generator in the training phase of the original model, extra processes are introduced in the training phase, which are supervision sample selection, noise training, and generator training. Therefore, we investigate the extra training time and report the mean training time cost of each part across all training epochs. According to the result in Figure 5, the noise training and generator training both take around 10% time cost in the training phase for all tasks on the MNIST style dataset. Since we set the generator training round as 200 for the task on CIFAR-10, the time cost of the generator training also doubled. Although the supervision sample selection (shown in orange) takes about 20% of the training time, it can be further reduced by reducing the sample selection times. For example, we can select supervision samples once for several epochs instead of selecting samples before every generator training.

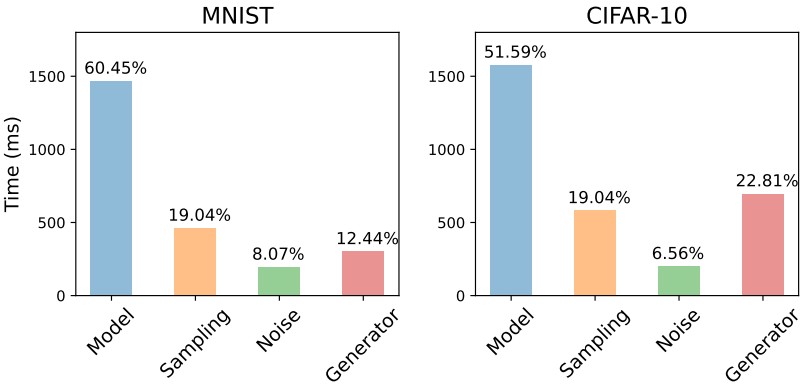

Figure 5: Time costs in training phase.

# F    EFFECT OF THE IMBALANCE RATE $r$

In this section, we examine the effect of the imbalance rate $r$, defined in Section 3. We also drop the assumption that all minority classes have a similar amount of data. By doing these two further investigations, we observe how these generative based unlearning methods perform with different levels of class imbalance. Specifically, we choose the imbalance rate $r$ in $[0.1, 0.2, 0.4]$ to simulate different levels of class imbalance. We also set $r$ to *"vary"* for the situation where minority classes have different amounts of data. When $r$ is set to *"vary"*, we randomly set the imbalance rate for each class, in this experience they are $[0.2, 0.7, 0.3, 0.3, 0.6, 0.2, 0.2, 0.6, 0.2, 0.6]$. For example, when it is going to unlearn the 1-th class, all data of the 1-th class in the training set will be kept to make it majority and only 20% of 0-th class data will be kept.

As shown in the Table 8, when the $r$ is grater than 0.1, the UNSIR outperforms the GENIU. This is because the UNSIR can access the retaining class data in the training set, and the amount of retaining class data in these cases is sufficient for the UNSIR to obtain enough information about the distribution of minority classes and repair its performance on these classes. However, the UNSIR will fail when the imbalance is more severe, for example when $r = 0.1$. Our proposed GENIU, which has no access to both forgetting and retaining class data in the unlearning phase, demonstrates its effectiveness in any imbalanced situation.

Table 8: Effect of the imbalance rate $r$.

| $r$ | Acc | origin | retrain | GKT | UNSIR | GENIU |
|-----|-----|--------|---------|-----|-------|-------|
| 0.1 | $\mathcal{D}_r \uparrow$ | 0.8057 | 0.816 | 0.2595 | 0.3002 | 0.7711 |
|     | $\mathcal{D}_f \downarrow$ | 0.9681 | 0.0 | 0.0 | 0.0016 | 0.0002 |
| 0.2 | $\mathcal{D}_r \uparrow$ | 0.8591 | 0.8693 | 0.2593 | 0.7871 | 0.7509 |
|     | $\mathcal{D}_f \downarrow$ | 0.954 | 0.0 | 0.0 | 0.0163 | 0.0 |
| 0.4 | $\mathcal{D}_r \uparrow$ | 0.8908 | 0.9013 | 0.2241 | 0.8742 | 0.7706 |
|     | $\mathcal{D}_f \downarrow$ | 0.9411 | 0.0 | 0.0 | 0.0031 | 0.0 |
| Vary | $\mathcal{D}_r \uparrow$ | 0.9022 | 0.9146 | 0.2643 | 0.8942 | 0.7992 |
|     | $\mathcal{D}_f \downarrow$ | 0.9375 | 0.0 | 0.0 | 0.0123 | 0.0001 |

# G    ABLATION STUDY

## G.1    MAIN CONTRIBUTION ABLATION.

In this section, we investigate the impact of two main technical components on unlearning performance, those are 1) the training proxy generator along with the training of the original model and 2) the in-batch tuning in unlearning phase. Detailedly, we compare the former with the post-training generated proxy and compare the latter with the widely used impair-repair process. For both impair and repair, we set the learning rate and the number of rounds as the same as in-batch tuning in the GENIU. From the results which are reported in Table. 9, when the proxy generated by the GENIU framework is applied, the impair-repair process will first forget the knowledge related to the majority class, however, this part of knowledge is most of the knowledge of the model about the classification task and makes the model hard to maintain the performance on retain classes in the subsequent repair stage. When the post-training generated proxy is used, due to the imbalance of original training data, the proxy of other classes can also present the characteristics of the majority class, thus, the ability to discriminate retain classes is also reduced after the majority class is forgotten.

Table 9: Main contributions ablation

| Proxy | Tuning | $Acc_u$ | $Acc_r$ |
|-------|--------|---------|---------|
| Post | Impair-Repair | 0.123 | 0.27 |
| GENIU | Impair-Repair | 0.048 | 0.758 |
| Post | GENIU | 0.018 | 0.416 |
| GENIU | GENIU | 0.0 | 0.771 |

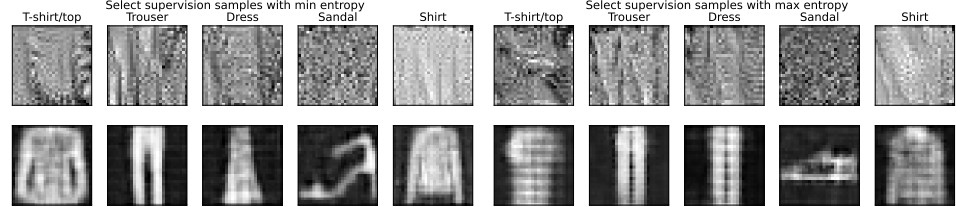

Figure 6: The figure shows the learned noises (upper) and corresponding generated proxies (lower) of some classes in the Fashion-MNIST dataset under different supervision sample selection conditions.

Table 10: Performance comparison between different type of supervision sample selection. (Fashion-MNIST)

| Selection | Acc | T-shirt | Trouser | Pullover | Dress | Coat | Sandal | Shirt | Sneaker | Bag | Boot |
|---|---|---|---|---|---|---|---|---|---|---|---|
| Max | $\mathcal{D}_r \uparrow$ | **0.784** | **0.743** | **0.743** | **0.799** | **0.829** | **0.7** | 0.79 | **0.753** | **0.775** | **0.796** |
| | $\mathcal{D}_u \downarrow$ | 0.0 | 0.0 | 0.0 | 0.0 | 0.0 | 0.0 | 0.0 | 0.0 | 0.0 | 0.002 |
| Min | $\mathcal{D}_r \uparrow$ | 0.78 | 0.68 | 0.721 | 0.775 | 0.768 | 0.648 | **0.844** | 0.64 | 0.755 | 0.751 |
| | $\mathcal{D}_u \downarrow$ | 0.027 | 0.0 | 0.001 | 0.0 | 0.0 | 0.0 | 0.0 | 0.0 | 0.008 | 0.0 |

## G.2 SUPERVISION SAMPLE SELECTION.

In this section, we will take the Fashion-MNIST dataset as an example to compare the impact of different supervision sample selection methods on unlearning performance. Specifically, in 4.4 we select $x$ with maximum logits entropy for each class. Therefore, in this ablation study, we compare the performance of selecting $x$ with maximum logits entropy and selecting with minimum logits entropy.

From Figure 6, we can see that the proxies trained by the max entropy sample are more visually blurred than the proxies trained by the min entropy sample. In addition, semantically, some skirts that look like pants will be selected, and some sandals that look like sneakers will also be selected. We intend to strengthen the model's discrimination of categories through these proxies with relatively high classification uncertainty.

From the results in Table. 10, we can see that, generally, when we use the maximum logits entropy method to select the supervision sample for the generator, the performance of unlearning will be better than using the minimum logits entropy method.

## G.3 IN-BATCH TUNING ROUNDS.

In this section, we conduct an ablation experiment to observe the effect of the number of rounds using in-batch tuning (a.k.a. unlearning rounds) on unlearning performance. Specifically, we continuously recorded the *accuracy* and *test error* of the $f(\cdot, \theta_{un})$ on unlearn test data and retain test data after each round in the unlearning process. From Figure 7, we can observe that in the first few rounds of unlearning, the difference between unlearn error and the retrain error is small, and the retain accuracy will also decrease slightly when the unlearn accuracy decreases. As the gap between

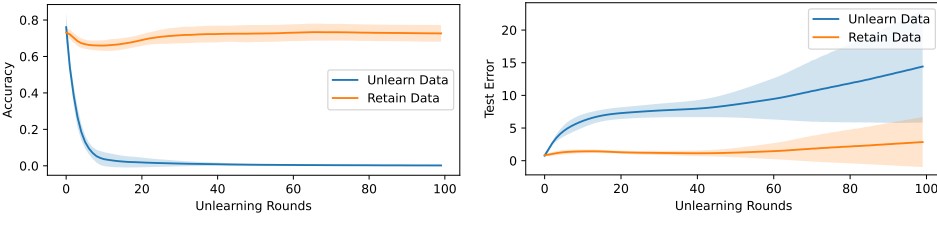

Figure 7: Impact of unlearning rounds.

unlearn error and retrain error increases, unlearn accuracy quickly drops to zero, while retain accuracy will gradually increase and gradually stabilize. Therefore, in the previous experiment, for the case where the accuracy of the $f(\cdot, \theta_{un})$ on $\mathcal{D}_f$ is small but not zero, it can be eliminated by increasing the unlearning rounds.

### G.4  TIME AND STORAGE COST OF AUXILIARY MODELS.

As we place the training of the proxy generator in the training phase of the original model, extra processes are introduced in the training phase, which are supervision sample selection, noise training, and generator training. Therefore, we investigate the extra training time and report the mean training time cost of each part across all training epochs. According to the result in Figure 5 (Appendix E), the noise training and generator training both take around 10% time cost in the training phase for all tasks on the MNIST style dataset. Since we set the generator training round as 200 for the task on CIFAR-10, the time cost of the generator training also doubled. Although the supervision sample selection (shown in orange) takes about 20% of the training time, it can be further reduced by reducing the sample selection times. For example, we can select supervision samples once for several epochs instead of selecting samples before every generator training.

For the storage cost, if still keeping the original data in the storage, the MNIST style dataset requires 45MB space and the CIFAR-10 requires 169MB. However, if saving the generator instead of original data, the generator requires only 4.6MB for MNIST's and 6.1MB for the CIFAR-10.

### G.5  START THE NOISE TRAINING WHEN THE ORIGINAL MODEL HAS DIFFERENT EFFECTIVENESS.

We conducted this experiment on the F-MNIST dataset. We define a classification accuracy threshold $t$ to determine when to start the noise training in the training phase. Since the original model can achieve 0.8 accuracy on the retaining classes, we set $t \in [0.4, 0.6, 0.7]$.

According to the results, we can see that as the threshold $t$ increases, the performance of the unlearned model will decrease accordingly. This is because the later the noise training is started, the more deeply the original model is affected by imbalanced data, and the worse the noise trained through original model is.

| Acc to start the noise training | Acc on $\mathcal{D}_r$ ↑ | Acc on $\mathcal{D}_f$ ↓ |
|---|---|---|
| none | 0.7711 | 0.0002 |
| 0.4 | 0.7705 | 0.0002 |
| 0.6 | 0.7544 | 0.0023 |
| 0.7 | 0.7436 | 0.0031 |

Table 11: Start the noise training when the original model has different effectiveness.

### G.6  THE SIZE OF MINI-BATCH

The mini-batch is actually the proxy set $\mathcal{D}_p$ in the problem definition (Section 3). Each element in $\mathcal{D}_p$, i.e. each proxy, is learned and generated using the maximum decision entropy sample as a reference (Section G.2). Therefore, these proxies can be considered as examples of each class that are very close to the decision boundary. In the unlearning phase, tuning the model with the information provided by such proxies can modify the decision boundary of the model as significantly as possible.

We did some additional experiments to have a quick look at the effect of different numbers of such mini-batches, i.e. different sizes of $\mathcal{D}_p$, on the unlearning results. We create a mini-batch containing one sample for each class. Then we set the different number of batches to run the experiment. In this experimental setting, there will be multiple proxies for a class. In order to make the proxies diverse, we will choose top-$B$ samples with maximum decision entropy as their supervisory information respectively, where $B$ is the number of the mini-batch.

As can be seen from the Table 12, the performance of unlearning will first increase and then decrease as the number of batches increases, which means that more proxies are not better. A reasonable

number of proxies will increase the diversity and information richness of $\mathcal{D}_p$ and help to better modify the decision boundary. However, if a larger number of proxies is required, due to the single and insufficient diversity of the training of the noise that is used to guide the proxy generation, the diversity of the learning of the generator is limited and the performance of unlearning is reduced. This may be a shortcoming of GENIU at this stage and is also one of our future work to improve the GENIU.

| $B$ | Acc on $\mathcal{D}_r$ ↑ | Acc on $\mathcal{D}_f$ ↓ |
|-----|-----|-----|
| 1 | 0.7711 | 0.0002 |
| 2 | 0.7825 | 0.0000 |
| 5 | 0.8004 | 0.0001 |
| 7 | 0.7953 | 0.0003 |
| 10 | 0.7971 | 0.0000 |
| 12 | 0.7693 | 0.0013 |

Table 12: Unlearning using different number of mini-batch.

