# OpenReview forum: "GENIU: A Restricted Data Access Unlearning for Imbalanced Data"
_ICLR.cc/2024/Conference — Submitted to ICLR 2024_

### Official Review · Reviewer_PTAU · 2023-10-30

**Soundness:** 3 good
**Presentation:** 3 good
**Contribution:** 3 good
**Rating:** 8
**Confidence:** 4

**Summary:**

This paper presents a novel approach to address the problem of class unlearning under a crucial challenge in learning from imbalanced data with restricted data access. Unlike traditional retraining methods that rely on the original data, this study introduces a special proxy generator method and an in-batch tuning strategy tailored for scenarios where the forgetting data predominantly consists of majority class samples. The proposed generator employs a generative approach to create a limited set of proxies for each class, effectively mitigating the impact of imbalanced data on the representation of minority classes. Additionally, the method introduces privacy-preserving measures by using noisy samples as generator seeds rather than relying on original training data. The empirical results demonstrate superior performance in both efficiency and effectiveness, showcasing the method's potential for practical applications.

**Strengths:**

This paper introduces a novel approach to address the challenging problem of imbalanced class unlearning with restricted data access. Unlike conventional retraining methods, the proposed special proxy generator method and in-batch tuning strategy offer a new perspective on efficiently unlearning from imbalanced data, particularly when forgetting data is predominantly composed of majority class samples. The paper's innovative use of a generative approach for proxy generation, as well as the integration of Variational Autoencoders (VAE) and in-batch tuning, showcases a creative combination of existing ideas, pushing the boundaries of current knowledge in this domain.
The empirical results of this study demonstrate excellent quality in terms of both efficiency and effectiveness. The proposed method exhibits superior performance, effectively mitigating the impact of imbalanced data on the representation of minority classes. The rigorous experimental validation underscores the robustness and reliability of the proposed approach, ensuring its applicability in real-world settings. Additionally, the introduction of privacy-preserving measures with noisy samples as generator seeds adds an extra layer of quality assurance.
The paper addresses a critical challenge in learning from imbalanced data with restricted data access. By focusing on class unlearning, the study not only defines a specific problem formulation but also offers a solution that has potential implications for various domains and applications. The introduction of a method that does not rely on original training data, regardless of forgetting or retaining it, marks a substantial advancement in the field. Furthermore, the efficient use of storage and time resources enhances the practical relevance and impact of the proposed method. The comprehensive empirical studies provided in the paper further solidify its motivation and validate the superiority of the proposed methods over existing approaches.

**Weaknesses:**

There are clarity issues in the technical details and notation definitions. Some related work discussions also lack clarity in illustrating how the challenges posed by the problem studied in this paper specifically affect those approaches. Furthermore, the assumption of similar data volumes for all minority classes requires clarification. It remains unclear whether this assumption is driven by the criticality of the imbalance rate as a hyperparameter in this paper.  These clarity issues collectively pose potential barriers to a comprehensive understanding of the paper's technical content. (See details in "Questions")

Additionally, there are some typos, such as "with the retraining of a new model" instead of "retaining of a new model." In the section discussing learning and unlearning from imbalanced data, the phrase "some shards may ..." could benefit from a more precise expression like "some shards may be composed of."

Another aspect to consider is that the proposed method entails generator training during the overall training process rather than relying on a pre-trained model. While this may impose certain limitations on the method's applicability, this unique approach also plays a crucial role in addressing the imbalance issue during generator training.

**Questions:**

1) The rationale behind assuming that all minority classes have a comparable volume of data warrants further clarification. Is this assumption driven by the significance of the imbalance rate as a critical hyperparameter in this paper? Moreover, it would be beneficial to address what implications arise if the minority classes do not exhibit a similar distribution of data. Providing
2) Why the knowledge of the generator cannot be accurately obtained in the unlearning phase? (at the beginning of 2nd paragraph, Sec 4.1)
3) \sigma and \mu in Eq 5 is not defined in the main text but in the title of Figure 2. Are they the parameters of the gaussian distribution?
4) What is D_s at the end of Sec 4.4?

---

> ### Author Response · Authors · 2023-11-16
> **Response to Reviewe PTAU (Part 1/2)**
>
> We would like to express our sincere gratitude for your thorough review of our manuscript, your constructive criticism and insightful questions. We deeply appreciate your positive feedback on our novel approach, particularly your recognition of our method's creative use of VAE, and the practical significance of our empirical results. We will rectified the typos and ambiguities in expression you pointed out to ensure that the revised manuscript offers a clearer, more comprehensive understanding of our methods and findings. Below we address your questions in detail and also provide clarify for some of the issues pointed out in the weakness part.
>
> ### **Q1: About the assumption that all minority classes have a comparable volume of data**
>
> We assumed a comparable volume of data across all minority classes primarily to simplify the description and ensure a consistent approach in our experiments. This assumption allowed us to use a single imbalance rate $r$ as a control variable, thus maintaining the focus on our method's efficacy under controlled conditions.
>
> Inspired by the question, we conducted additional tests to assess the performance of our method when dealing with varying data sizes across different minority classes. Specifically, we maintain a constant size for the majority class while varying the size of the minority class data by adjusting the imbalance rate, denoted as $r$. A smaller value of $r$ corresponds to a reduced size of the minority data, thereby increasing the level of imbalance in the dataset. These supplementary experiments (shown below and details will be included in the revised paper Appendix F) demonstrate that our proposed method consistently outperforms baseline approaches. This further validates the robustness and adaptability of our approach in handling diverse real-world scenarios.
>
> Below we provide a detailed description of the supplementary experiments and their results:
> We examine the effect of the imbalance rate $r$, defined in Section 3. We also drop the assumption that all minority classes have a similar amount of data. By doing these two further investigations, we observe how these generative based unlearning methods perform with different levels of class imbalance. Specifically, we choose the imbalance rate $r$ in $[0.1, 0.2, 0.4]$ to simulate different levels of class imbalance. We also set $r$ to ***''vary''*** for the situation where minority classes have different amounts of data. When $r$ is set to ***''vary''***, we randomly set the imbalance rate for each class, in this experience they are $[0.2, 0.7, 0.3, 0.3, 0.6, 0.2, 0.2, 0.6, 0.2, 0.6]$. For example, when it is going to unlearn the 1-th class, all data of the 1-th class in the training set will be kept to make it majority and only 20\% of 0-th class data will be kept.
>
> As shown in the table, when the $r$ is grater than 0.1, the UNSIR outperforms the GENIU. This is because the UNSIR can access the retaining class data in the training set, and the amount of retaining class data in these cases is sufficient for the UNSIR to obtain enough information about the distribution of minority classes and repair its performance on these classes. However, the UNSIR will fail when the imbalance is more severe, for example when $r=0.1$. Our proposed GENIU, which has no access to both forgetting and retaining class data in the unlearning phase, demonstrates its effectiveness in any imbalanced situation.
>
> | r     | Acc                | origin | retrain | GKT    | UNSIR  | GENIU    |
> |-------|--------------------|--------|---------|--------|--------|--------|
> | 0.1   | $\mathcal{D}_r\uparrow$ | 0.8057 | 0.816   | 0.2595 | 0.3002 | 0.7711 |
> |       | $\mathcal{D}_f\downarrow$ | 0.9681 | 0.0     | 0.0    | 0.0016 | 0.0002 |
> | 0.2   | $\mathcal{D}_r\uparrow$ | 0.8591 | 0.8693  | 0.2593 | 0.7871 | 0.7509 |
> |       | $\mathcal{D}_f\downarrow$ | 0.954  | 0.0     | 0.0    | 0.0163 | 0.0    |
> | 0.4   | $\mathcal{D}_r\uparrow$ | 0.8908 | 0.9013  | 0.2241 | 0.8742 | 0.7706 |
> |       | $\mathcal{D}_f\downarrow$ | 0.9411 | 0.0     | 0.0    | 0.0031 | 0.0    |
> | Vary  | $\mathcal{D}_r\uparrow$ | 0.9022 | 0.9146  | 0.2643 | 0.8942 | 0.7992 |
> |       | $\mathcal{D}_f\downarrow$ | 0.9375 | 0.0     | 0.0    | 0.0123 | 0.0001 |

---

> ### Author Response · Authors · 2023-11-16
> **Response to Reviewe PTAU (Part 2/2)**
>
> ### **Q2: Why the knowledge of the generator cannot be accurately obtained in the unlearning phase? (at the beginning of 2nd paragraph, Sec 4.1)**
>
> In the unlearning phase, the accurate knowledge of the generator cannot be obtained due to the absence of training data. This limitation stems from our model's design to adhere to restricted data access conditions. Since we only have access to the trained classifier and not the training data during this phase, it becomes infeasible to obtain a generator that can accurately capture and describe the data characteristics of minority class as the knowledge of the classifier itself may be biased due to imbalanced data. This constraint is a critical aspect of our approach, focusing on unlearning under privacy-preserving conditions where access to the original training data is not possible.
>
> ### **Q3: \sigma and \mu in Eq 5 is not defined in the main text but in the title of Figure 2. Are they the parameters of the gaussian distribution?**
>
> Yes, these are the parameters of the Gaussian distribution. We have revised this in our paper in Section 4.1 to make it clear and the revised paper will be uploaded soon.
>
> ### **Q4: What is D_s at the end of Sec 4.4**
>
> $D_s$ represents a set of selected samples from the training dataset, specifically chosen for training the generator. These samples are strategically sampled to effectively train the generator, ensuring it accurately captures the key characteristics and distributions of the training data. We have revised our paper in Section 4.4 to make it clear and the revised paper will be uploaded soon.
>
>
> **In addtion, we appreciate your insightful comments regarding the clarity issues in our paper and the identification of specific typos. In response to your feedback:**
>
> 1. **Technical Details and Notation Definitions**: We have carefully revised the technical sections to enhance clarity. This includes a more explicit explanation of notation and definitions, ensuring that our approach and its uniqueness are clearly communicated. Specifically, we enhance clarity of our problem definition at the end of Section 3 and the description of our method in Section 4.1.
>
> 2. **Discussion of Related Work**: We have expanded the discussion on related work, particularly focusing on how the challenges posed by our study affect these approaches ("Data restricted unlearning methods" in Section 2). This revision aims to provide a more comprehensive understanding of our work in the context of existing literature.
>
> 3. **Correction of Typos**: We have corrected the identified typos, such as replacing "retraining of a new model" with "retaining of a new model" and refining the expression in the section on learning and unlearning from imbalanced data. These corrections are aimed at improving the precision and readability of our paper.
>
> We have revised the paper accordingly and the revised version will be uploaded soon.
>
> We believe these revisions significantly enhance the paper's quality and address the concerns you raised. We are committed to presenting our research as clearly and accurately as possible, and we are appreciate your reviews in achieving this goal. We are also grateful for your insightful suggestion regarding the use of a pre-trained model for generator training in our method. Initially, this aspect was not a primary consideration in our research design. However, your insights have highlighted a promising direction for future improvement of our GENIU framework. Thank you again for your valuable feedback.

---

> ### Author Response · Authors · 2023-11-20
> **Looking forward to your further suggestions**
>
> Dear Reviewer PTAU, we greatly appreciate your recognition of our work and your valuable suggestions. Based on your suggestions, we will improve the wording of the paper. In response to your questions about the degree of imbalance, we have conducted additional experiments and the results and analyses are also included in the response. We would like to thank you for your time and review comments and look forward to hearing your latest suggestions and welcome further discussion.

---

> ### Author Response · Authors · 2023-11-22
> **The latest revision has been uploaded.**
>
> Dear reviewers, I would like to thank you once again for your valuable review comments, which are very helpful in improving our work. We have carefully referred your comments and revised the paper accordingly, and the latest revised version has been uploaded. We would really appreciate the opportunity to receive your latest comments and discuss further before the discussion period closes. Much appreciate.

---

### Official Review · Reviewer_iMwM · 2023-10-31

**Soundness:** 3 good
**Presentation:** 4 excellent
**Contribution:** 2 fair
**Rating:** 5
**Confidence:** 3

**Summary:**

This paper proposes a novel approach to class unlearning for imbalanced data called GENIU. The framework utilizes a Variational Autoencoder (VAE) to generate accurate proxies for the forgetting data, even when it consists predominantly of the majority class. The proposed GENIU is divided into training phase and unlearning phase. During the training phase, the VAE is trained to generate accurate proxies for the forgetting data. During the unlearning phase, the VAE is used to generate proxies for the forgetting data, which are then used to update the model. The paper's contributions include a detailed description of the GENIU algorithm, an evaluation of its performance on several benchmark datasets, and a discussion of its potential applications and future research directions.

**Strengths:**

- This paper is well-written and easy for readers to understand, and its key idea is clear.

- This paper deals with scenarios that could occur in the real world, such as situations where access to original data is not possible or where a classification model is trained on imbalanced data. These are plausible constraints, and the paper provides sufficient approaches to address them.

- The authors conducted sufficient experiments to explain their algorithms and also conducted a thorough analysis of it. They provide examples such as explaining the storage space advantage, unlearning for various classes, and multi-class unlearning.

**Weaknesses:**

- The authors train a generative model together with a classification model at the beginning. However, this can be a critical privacy issue because the generative model itself contains information about the forgetting data. Although this can be discarded after one unlearning process, it cannot be used for the next unlearning process. Therefore, it seems to be an architecture that cannot perform continuous unlearning.

- The generated proxies are mentioned to be far from the decision boundary. In other words, the model successfully classifies them with high confidence. However, the statement that high-entropy data is selected for learning this contradicts this idea. Shouldn't the opposite data be selected for this? The explanation for these aspects is more clearly explained.

- The paper does not address how GENIU scales with significantly larger datasets or more complex models. This leaves questions about its feasibility in highly demanding real-world scenarios. It would be better to provide a more detailed comparative analysis, especially highlighting specific scenarios where GENIU might not be the optimal choice.

- There are sentences that appear to be grammar errors and typos. Examples include "~ imbalanced data, if train the proxies throught the ~" in "4. our method", "ALLCNN 20 epochs ~ noise z the" in "5. implementation detail", and "unmber" in "algorithm 1". The authors seem to need to pay more attention to these errors.

**Questions:**

- I can understand that the method of generating proxies using a well-trained model as a guide contains many characteristics of the majority class. However, it seems that there is a lack of analysis on how the approach of training both the classification model and the generation model together addresses the issue of imbalanced characteristics. Ultimately, it appears that the authors' idea is correct, but I'm curious about why that is the case.

- During the training process, learning from noisy images is also carried out. Isn't it valid to do this when the performance of the classification model is reasonably assured? Isn't it possible to save noisy images that happened to match when the performance is low?

- I'm wondering whether the authors have considered a method involving the use of a negative value in the formulation of the loss function for forgetting data, as they took the reciprocal.

- Is one mini-batch sufficient? Why is that?

- Could you provide more insight into the assumption that all minority classes have similar data counts? How might GENIU perform if faced with varying levels of class imbalance within a dataset?

---

> ### Author Response · Authors · 2023-11-16
> **Response to Reviewer iMwM (Part 1/6)**
>
> Thank you for your valuable and detailed feedback on our paper. We are grateful for your positive comments on the clarity, real-world applicability, and thorough experimentation of our paper. We appreciate the opportunity to clarify and address the concerns you have raised regarding its weaknesses and questions below.
>
> ### **Q1: How does the approach of training both the classification model and the generation model together addresses the issue of imbalanced characteristics**
>
> Thank you for your insightful question. Our approach is grounded in a practical intuition: employing an auxiliary network (the generator) to focus on equitably capturing sample features across classes.
>
> This generator, built on a Variational Autoencoder (VAE) structure, is uniquely tasked with capturing class feature information, independent of the classification task. VAEs excel in projecting data onto a more compact, yet sufficiently discriminative, low-dimensional representation. This enhances computational efficiency, as detailed in the works of Kingma and Welling (2013) and Pham and Le (2020) [1,2].
>
> To mitigate the impact of data imbalance, the generator is trained with a balanced data sampling approach (refer to Section 4.4). This balanced training is crucial for the generator to accurately capture class-specific features. The concurrent training of the classification and generation models is strategically chosen to allow the generator to learn directly from actual samples available during the training phase. This approach not only facilitates the acquisition of genuine sample characteristics but also optimizes the process by reducing IO time overhead.
>
> While the explanations are not backed by rigorous mathematical proofs, our ablation study in Section 5.2 supports this intuition. The experimental results (see Table 5) indicate that training the generator separately from the original model $f(\cdot,\theta _{or})$ significantly diminishes GENIU’s performance. Hence, accurately capturing class features when actual samples are accessible is pivotal for generating effective proxies in scenarios plagued by class imbalance.
>
> **References:**
>
> [1] Kingma, Diederik P., and Max Welling. "Auto-Encoding Variational Bayes." arXiv preprint arXiv:1312.6114 (2013).
>
> [2] Pham, Dang, and Tuan Le. "Auto-Encoding Variational Bayes for Inferring Topics and Visualization." arXiv preprint arXiv:2010.09233 (2020).

---

> ### Author Response · Authors · 2023-11-16
> **Response to Reviewer iMwM (Part 2/6)**
>
> ### **Q2: During the training process, learning from noisy images is also carried out. Isn't it valid to do this when the performance of the classification model is reasonably assured? Isn't it possible to save noisy images that happened to match when the performance is low?**
>
> Thank you for posing this intuitive and important question. We initially shared the same expectation that introducing noisy images later in the training process — once the classification model's performance stabilizes — would yield better results. However, our experiments with the F-MNIST dataset presented a surprising trend that challenges this expectation.
>
> In these experiments, we varied the timing of noisy image introduction based on different accuracy thresholds: 0.4, 0.6, and 0.7. Contrary to our expectations, we found that the later we introduced noisy images, the less effectively the model performed in the unlearning process. The results below illustrate this trend:
>
> | Threshold for Starting Noise Training | Accuracy on $\mathcal{D}_r\uparrow$ | Accuracy on $\mathcal{D}_f\downarrow$ |
> |---------------------------------------|-------------------------------------|---------------------------------------|
> | None (baseline)                       | 0.7711                              | 0.0002                                |
> | 0.4                                   | 0.7705                              | 0.0002                                |
> | 0.6                                   | 0.7544                              | 0.0023                                |
> | 0.7                                   | 0.7436                              | 0.0031                                |
>
> We think that this phenomenon is linked to how the model adapts to imbalanced data. When noisy images are introduced at a later stage, the model has already significantly adapted to the majority class. This makes it more challenging for the model to adapt to and "unlearn" from the newly introduced noise, reducing the effectiveness of the unlearning process.
>
> Thus, our findings suggest that introducing noisy images earlier in the training process, even before high accuracy is achieved, might be more conducive to effective unlearning in scenarios with imbalanced data.
>
>
> ### **Q3: About the use of a negative value in the formulation of the loss function for forgetting data**
>
> Thank you for your query regarding the use of a negative value in the loss function for forgetting data. We thoroughly considered this approach and concluded that employing a negative value could lead to instability in the learning process. Specifically, as learning progresses, using a negative value for the forgetting loss risks causing divergence. This divergence can lead to a scenario where the absolute value of the forgetting loss significantly outweighs that of the retaining loss, undermining the model's ability to retain knowledge of the retaining classes. This concern aligns with discussions in other work, such as Poppi et al. (2023) [1].
>
> To empirically demonstrate this, we conducted an additional experiment comparing the use of a negative loss with the reciprocal approach. The results, as shown in the tables below, indicate that a negative loss leads to divergence in loss values, resulting in unsuccessful unlearning:
>
> **Loss Value in Epoch for Negative Loss**
>
> | Epoch | Loss Value |
> |-------|------------|
> | 5     | -0.3929    |
> | 10    | -1.479     |
> | 20    | -7.4278    |
> | 40    | -194.7163  |
> | 80    | -240270.2717 |
>
> **Accuracy for Negative and Reciprocal Loss**
>
> | Loss Type  | Acc on $\mathcal{D}_f\downarrow$ | Acc on $\mathcal{D}_r\uparrow$ |
> |------------|----------------------------------|--------------------------------|
> | Negative   | 0.0191                           | 0.1944                         |
> | Reciprocal | 0.0002                           | 0.7711                         |
>
> In contrast, the reciprocal approach maintains stability and effectiveness in unlearning, as evidenced by the consistent loss values and better performance on both $\mathcal{D}_f$ and $\mathcal{D}_r$ datasets.
>
> Reference:
> [1] Poppi, Samuele, et al. "Multi-Class Explainable Unlearning for Image Classification via Weight Filtering." arXiv preprint arXiv:2304.02049 (2023).

---

> ### Author Response · Authors · 2023-11-16
> **Response to Reviewer iMwM (Part 3/6)**
>
> ### **Q4: Is one mini-batch sufficient? Why is that?**
>
> Thank you for your valuable suggestion. In response, we conducted additional experiments to explore the impact of number of mini-batches and assess its potential in enhancing GENIU.
>
> Specifically, we varied the number of batches. To ensure diversity among proxies, we selected the top-$B$ samples with maximum decision entropy, where $B$ represents the batch count.
>
> Our experimental findings, summarized in the table below, reveal an interesting trend: the performance of unlearning initially improves with an increase in batch numbers, then begins to decline. This indicates that while a moderate number of proxies enhances the diversity and informational richness, leading to more effective decision boundary modification, an excessive number reduces performance.
>
> This observed reduction suggests an important direction for future enhancement of GENIU. It indicates that expanding the diversity in the noise training, which guides proxy generation, could further improve the model's performance.
>
> | $B$ | Acc on $\mathcal{D}_r\uparrow$ | Acc on $\mathcal{D}_f\downarrow$ |
> |-----|-------------------------------|----------------------------------|
> | 1   | 0.7711                        | 0.0002                           |
> | 2   | 0.7825                        | 0.0000                           |
> | 5   | 0.8004                        | 0.0001                           |
> | 7   | 0.7953                        | 0.0003                           |
> | 10  | 0.7971                        | 0.0000                           |
> | 12  | 0.7693                        | 0.0013                           |

---

> ### Author Response · Authors · 2023-11-16
> **Response to Reviewer iMwM (Part 4/6)**
>
> ### **Q5: About the assumption that all minority classes have a comparable volume of data**
>
> We assumed a comparable volume of data across all minority classes primarily to simplify the description and ensure a consistent approach in our experiments. This assumption allowed us to use a single imbalance rate $r$ as a control variable, thus maintaining the focus on our method's efficacy under controlled conditions.
>
> Inspired by the question, we conducted additional tests to assess the performance of our method when dealing with varying data sizes across different minority classes. Specifically, we maintain a constant size for the majority class while varying the size of the minority class data by adjusting the imbalance rate, denoted as $r$. A smaller value of $r$ corresponds to a reduced size of the minority data, thereby increasing the level of imbalance in the dataset. These supplementary experiments (shown below and details will be included in the revised paper Appendix F) demonstrate that our proposed method consistently outperforms baseline approaches. This further validates the robustness and adaptability of our approach in handling diverse real-world scenarios.
>
> Below we provide a detailed description of the supplementary experiments and their results:
> We examine the effect of the imbalance rate $r$, defined in Section 3. We also drop the assumption that all minority classes have a similar amount of data. By doing these two further investigations, we observe how these generative based unlearning methods perform with different levels of class imbalance. Specifically, we choose the imbalance rate $r$ in $[0.1, 0.2, 0.4]$ to simulate different levels of class imbalance. We also set $r$ to ***''vary''*** for the situation where minority classes have different amounts of data. When $r$ is set to ***''vary''***, we randomly set the imbalance rate for each class, in this experience they are $[0.2, 0.7, 0.3, 0.3, 0.6, 0.2, 0.2, 0.6, 0.2, 0.6]$. For example, when it is going to unlearn the 1-th class, all data of the 1-th class in the training set will be kept to make it majority and only 20\% of 0-th class data will be kept.
>
> As shown in the table, when the $r$ is grater than 0.1, the UNSIR outperforms the GENIU. This is because the UNSIR can access the retaining class data in the training set, and the amount of retaining class data in these cases is sufficient for the UNSIR to obtain enough information about the distribution of minority classes and repair its performance on these classes. However, the UNSIR will fail when the imbalance is more severe, for example when $r=0.1$. Our proposed GENIU, which has no access to both forgetting and retaining class data in the unlearning phase, demonstrates its effectiveness in any imbalanced situation.
>
> | r     | Acc                | origin | retrain | GKT    | UNSIR  | GENIU    |
> |-------|--------------------|--------|---------|--------|--------|--------|
> | 0.1   | $\mathcal{D}_r\uparrow$ | 0.8057 | 0.816   | 0.2595 | 0.3002 | 0.7711 |
> |       | $\mathcal{D}_f\downarrow$ | 0.9681 | 0.0     | 0.0    | 0.0016 | 0.0002 |
> | 0.2   | $\mathcal{D}_r\uparrow$ | 0.8591 | 0.8693  | 0.2593 | 0.7871 | 0.7509 |
> |       | $\mathcal{D}_f\downarrow$ | 0.954  | 0.0     | 0.0    | 0.0163 | 0.0    |
> | 0.4   | $\mathcal{D}_r\uparrow$ | 0.8908 | 0.9013  | 0.2241 | 0.8742 | 0.7706 |
> |       | $\mathcal{D}_f\downarrow$ | 0.9411 | 0.0     | 0.0    | 0.0031 | 0.0    |
> | Vary  | $\mathcal{D}_r\uparrow$ | 0.9022 | 0.9146  | 0.2643 | 0.8942 | 0.7992 |
> |       | $\mathcal{D}_f\downarrow$ | 0.9375 | 0.0     | 0.0    | 0.0123 | 0.0001 |

---

> ### Author Response · Authors · 2023-11-16
> **Response to Reviewer iMwM (Part 5/6)**
>
> **Clarification for Comments in the Weaknesses Section**
>
> ### **W1: It seems to be an architecture that cannot perform continuous unlearning.**
>
> We appreciate your insights regarding the privacy implications and the challenge of continuous unlearning in our method.
>
> The design of our system is such that the forgetting data is class-specific. The generator relies on trained noise samples as prompts to generate proxies for the forgetting class. In the absence of corresponding noise, these proxies cannot be reconstructed, thus preventing the retrieval of forgetting information within our framework. This design inherently safeguards against inadvertent retrieval of forgotten data, addressing privacy concerns to some extent.
>
> However, we acknowledge the trade-off in our current approach: to enhance privacy preservation, we have had to compromise on the capability for continuous unlearning. This limitation mirrors challenges faced in continual learning, where models must adapt to new data while managing the retention and potential exposure of sensitive information over time. In continual learning environments, ensuring privacy becomes increasingly complex due to the evolving nature of the data and the need to retain relevant past knowledge without compromising individual privacy. This parallel highlights the inherent complexity of developing learning systems that are both dynamic and privacy-preserving.
>
> Thus, we recognize that continuous unlearning is an emerging topic in the field and holds significant importance, especially given its relevance in practical applications. This comment has highlighted a crucial area for further research and development. We believe as the field evolves, exploring methods that balance privacy preservation with the flexibility of continuous unlearning will be paramount, and we appreciate the opportunity provided here to consider these critical aspects and look forward to advancing our work in this direction.
>
> ### **W2: The generated proxies are mentioned to be far from the decision boundary. In other words, the model successfully classifies them with high confidence. However, the statement that high-entropy data is selected for learning this contradicts this idea. Shouldn't the opposite data be selected for this? The explanation for these aspects is more clearly explained.**
>
> We apologize for the confusion caused by the typo in our paper. We have made the necessary correction in Section 4.4 to clarify that the generated proxies are indeed intended to be close to the decision boundary, not far from it.

---

> ### Author Response · Authors · 2023-11-16
> **Response to Reviewer iMwM (Part 6/6)**
>
> ### **W3.1 The paper does not address how GENIU scales with significantly larger datasets or more complex models. This leaves questions about its feasibility in highly demanding real-world scenarios.**
>
> We appreciate your concern about the scalability of GENIU with larger datasets or more complex models, and its applicability in highly demanding real-world scenarios.
>
> At the current stage, while we have not extensively tested GENIU on significantly larger datasets, we have conducted supplementary experiments using a more complex network structure ResNet. These experiments provide initial insights into how GENIU performs with increased complexity compared to AllCNN.
>
> The following table summarizes our results:
>
> | Network | Metric                   | Original | Retrain | GKT    | UNSIR  | GENIU   |
> |---------|--------------------------|----------|---------|--------|--------|---------|
> | ResNet  | $\mathcal{D}_r\uparrow$   | 0.8218   | 0.8096  | 0.2229 | 0.2539 | 0.6437  |
> |         | $\mathcal{D}_f\downarrow$ | 0.9508   | 0.0     | 0.0    | 0.0    | 0.0     |
> | AllCNN  | $\mathcal{D}_r\uparrow$  | 0.8057   | 0.816   | 0.2595 | 0.3002 | 0.7711  |
> |         | $\mathcal{D}_f\downarrow$| 0.9681   | 0.0     | 0.0    | 0.0016 | 0.0002  |
>
> As indicated, GENIU maintains better performance compared to other methods even with the ResNet architecture, though there is a noted decrease in performance compared to when a simpler network structure is used. This suggests that while GENIU is adaptable to more complex models, further optimization might be necessary to fully leverage its capabilities in these contexts. Future work will focus on this aspect to ensure that GENIU remains effective and feasible in a variety of demanding real-world scenarios. This feedback is invaluable for guiding the next stages of our research.
>
> ### **W3.2 It would be better to provide a more detailed comparative analysis, especially highlighting specific scenarios where GENIU might not be the optimal choice.**
>
> We appreciate the suggestion to provide a more in-depth comparative analysis to highlight scenarios where our GENIU framework might not be the most suitable choice.
>
> One such scenario is when a portion, but not all, of the original classes of data are available during the unlearning phase. In these cases, GENIU may not be the most optimal solution. This is evidenced in our supplementary experiments  which has been listed below, where we observed that UNSIR outperforms GENIU under certain conditions. For example, when there is sufficient original data can be accessed by the UNSIR in the unlearning phase, the UNSIR can repair its performance well on the forgetting classes with actual samples.
>
> | r     | Acc                | origin | retrain | GKT    | UNSIR  | GENIU    |
> |-------|--------------------|--------|---------|--------|--------|--------|
> | 0.1   | $\mathcal{D}_r\uparrow$ | 0.8057 | 0.816   | 0.2595 | 0.3002 | 0.7711 |
> |       | $\mathcal{D}_f\downarrow$ | 0.9681 | 0.0     | 0.0    | 0.0016 | 0.0002 |
> | 0.2   | $\mathcal{D}_r\uparrow$ | 0.8591 | 0.8693  | 0.2593 | 0.7871 | 0.7509 |
> |       | $\mathcal{D}_f\downarrow$ | 0.954  | 0.0     | 0.0    | 0.0163 | 0.0    |
> | 0.4   | $\mathcal{D}_r\uparrow$ | 0.8908 | 0.9013  | 0.2241 | 0.8742 | 0.7706 |
> |       | $\mathcal{D}_f\downarrow$ | 0.9411 | 0.0     | 0.0    | 0.0031 | 0.0    |
> | Vary  | $\mathcal{D}_r\uparrow$ | 0.9022 | 0.9146  | 0.2643 | 0.8942 | 0.7992 |
> |       | $\mathcal{D}_f\downarrow$ | 0.9375 | 0.0     | 0.0    | 0.0123 | 0.0001 |
>
> We believe that these results and analysis are valuable for understanding the applicability and limitations of GENIU in real-world scenarios. And we will update our paper to include these results.
>
> ### **W4: There are sentences that appear to be grammar errors and typos. Examples include "~ imbalanced data, if train the proxies throught the ~" in "4. our method", "ALLCNN 20 epochs ~ noise z the" in "5. implementation detail", and "unmber" in "algorithm 1". The authors seem to need to pay more attention to these errors.**
>
> We deeply appreciate your thorough review and your attention to detail in identifying grammatical errors and typos in our manuscript. Following the reviews, we have meticulously reviewed and corrected the errors in the main text. The revised version will be uploaded soon. Thank you again for helping us improve the quality of our paper.

---

> ### Author Response · Authors · 2023-11-20
> **Looking forward to your further suggestions**
>
> Dear Reviewer iMwM, we sincerely appreciate your time and comprehensive review.  Following your valuable and insightful comments, we have added several experiments (including the concern of data size, loss function and more complicated network) and included the results and analysis in our response. We believe we have addressed your concerns in our response, and eagerly await your feedback on our response and hope to have the opportunity to answer any further questions you may have. Much appreciate.

---

> ### Author Response · Authors · 2023-11-22
> **The latest revision has been uploaded.**
>
> Dear reviewers, I would like to thank you once again for your valuable review comments, which are very helpful in improving our work. We have carefully referred your comments and revised the paper accordingly, and the latest revised version has been uploaded. We would really appreciate the opportunity to receive your latest comments and discuss further before the discussion period closes. Much appreciate.

---

### Official Review · Reviewer_iDvc · 2023-11-01

**Soundness:** 1 poor
**Presentation:** 2 fair
**Contribution:** 1 poor
**Rating:** 1
**Confidence:** 3

**Summary:**

This paper proposes a new method for approximate unlearning of in a class imbalanced setting with restricted data access., called GENIU. GENIU employs a VAE to simultaneously train a substitute generator in conjunction with the main model. The paper uses experiments to validate claims.

**Strengths:**

The clarity of the work is acceptable. Furthermore, the work is highly novel and original. The usage of VAE seems cool.

**Weaknesses:**

However, I find the following critical faults with the paper:

- The baselines seem ill-defined. In the presented experiments, there is not a good way of knowing what constitutes a good delta in classification based on an unlearning request. In the results table, the authors show that after unlearning, the accuracy for the unlearned class is 0.0. I do not understand why there is any merit in this. Throughout the entire paper, there is never any mention of what constitutes a valid "forgetting" of a given class (as an exact definition). It seems that the implied definition (based on the results) is that the accuracy on the forgotten class should be zero. I disagree that this is a useful definition.
- The formulation seems ill-defined. The authors do not do a precise job of describing the setting. For example: how imbalanced must the classes be for this method to work? How large of a fraction of unlearning can this method support?
- The motivation is unclear -- the authors fail to explain either practical or intellectual motivation for proposing this algorithm. I am left wondering why this setting matters.
- The method seems overly complicated --- the authors fail to include simpler methods and show that they fail.

**Questions:**

I would like to see:

1 - a stronger formulation section. This paper fails to defined the goals formally.
2 - more motivation
3 - better baselines

While the algorithm proposed is interesting, I feel there is significant work to be done before this is ready for publication.

---

> ### Author Response · Authors · 2023-11-16
> **Response to Reviewer iDvc (Part 1/6)**
>
> Thank you for your comprehensive review of our paper and for recognizing its "highly novel and original" aspects, as well as the "cool" usage of the VAE in our work. We greatly value your constructive feedback and are committed to enhance the clarity of our paper. In the following, we address each question you have raised in the Questions (Q) or Weaknesses (W) part.
>
> ### **Q1 and W2: A stronger formulation section. This paper fails to defined the goals formally.**
>
> Thank you for highlighting the need for a stronger formulation and the formal definition of our goals. Below, we provide a detailed formulation, step-by-step, leading to our goal, and we will also update Section 3 in the paper accordingly.
>
> - **The Learning Problem Formulation**: We define the learning problem as a $K$-class classification issue. Let $\mathcal{D}=\{(x_i,y_i)\}_{i=1}^{n} \in \mathcal{X}\times \mathcal{Y}$ represent a training set containing $n$ instances, where $x_i\in \mathcal{X}\subset \mathbb{R}^{d}$ and $y_i\in \mathcal{Y}=\{1,\dots,K\}$. Upon inputting the training data into a learning algorithm, such as a neural network, it produces a classifier $f(x,\theta)$, parameterized by $\theta$. This classifier takes an instance $x\in \mathcal{X}$ and outputs a vector ${p}\in [0,1]^K$, where the $j$th element of ${p}$ represents the posterior probability of the $j$th label given $x$, i.e., $p_j$ is interpreted as $P(y = j|x)$.
>
> - **The Class Unlearning Formulation**: The objective of class unlearning is to remove specific classes $\mathcal{Y}_f\subset \mathcal{Y}$ from a well-trained model $f(x, \theta)$. The ideal outcome of unlearning is to emulate a classifier, denoted as $f(x,\theta _{re})$, which is re-trained on retained data $D_r$ — data not associated with classes in $\mathcal{Y}_f$. Most class-unlearning algorithms aim to produce a classifier $f(x, \theta _{un})$ that approximates or matches the performance of $f(x,\theta _{re})$.
>
> - **The Restricted Data Access Formulation**: During the unlearning phase, there is no access to any element in the training data $\mathcal{D}$. Consequently, obtaining $f(x,\theta _{re})$ is impossible.
>
> - **The Imbalanced Data Formulation**: Let $\mathcal{D}^{k}=\{(x_i,y_i)|y_i = k\}$ be a subset of $\mathcal{D}$ containing training samples of the $k$-th class. In cases of data imbalance, the minority class data are significantly fewer than the majority class data, i.e., $|D^{k_1}| \ll |D^{k_2}|$, where $k_1$ and $k_2$ represent a minority and a majority class, respectively. Note that $\mathcal{Y}$ includes at least one majority class in this imbalanced setting.
>
> - **The Goal of Our Work**: Our aim is to develop an unlearning algorithm, $\mathcal{U}$, which can remove classes in $\mathcal{Y}_f$—containing at least one majority class—without needing access to any part of $\mathcal{D}$ during the unlearning phase. We anticipate that the output of $\mathcal{U}$, i.e., $f(x, \theta _{un})$, will closely approximate the performance of $f(x,\theta _{re})$, which assumes full access to the training data.

---

> ### Author Response · Authors · 2023-11-16
> **Response to Reviewer iDvc (Part 2/6)**
>
> ### **Q2 and W3: More motivation. The authors fail to explain either practical or intellectual motivation for proposing this algorithm. I am left wondering why this setting matters.**
>
> Thank you for emphasizing the importance of a stronger motivation in our paper. We recognize the necessity of explaining both the practical and intellectual motivations for proposing our algorithm, and will revise Sections 1 of our paper accordingly.
>
> **Practical Motivations:**
> 1. **Data Availability in Real-World Applications**: In many real-world scenarios, original training data may not always be available after training finishes due to privacy concerns or storage limitations. For example, sensitive data is often deleted post-training to protect privacy. Additionally, in streaming services, data is not stored long-term due to storage constraints. Our algorithm addresses these real-world challenges by enabling effective class unlearning without requiring access to the original data.
> 2. **Prevalence of Imbalanced Data**: Imbalanced data is a common issue in practical applications. Our work is motivated by the need to provide solutions for class unlearning in such imbalanced data scenarios.
>
> **Intellectual Motivation:**
> The intellectual motivation for our work is directly linked to the challenges posed by imbalanced data in the context of class unlearning. In such scenarios, the efficacy of existing generative methods, exemplified by GKT [1] and UNSIR [2], is often compromised due to their inability to accurately represent class characteristics when data is imbalanced. This problem becomes particularly pronounced in situations where access to original data is restricted, and unlearning relies heavily on the quality of proxy data generated by these methods. The prevalent imbalance in classes leads to generated proxies that fail to encapsulate the minority class adequately, thereby impacting the unlearning process. Our algorithm is designed to overcome these challenges, providing a more effective approach for class unlearning that is robust to the issues posed by imbalanced data settings.
>
> **Why This Setting Matters:**
> Addressing class unlearning in imbalanced data scenarios, particularly with restricted data access, represents a major, yet largely unexplored challenge in machine learning. Contrary to existing methods, which assume balanced data and complete access to original datasets, our work pioneers in identifying and tackling this critical research gap. Real-world applications frequently face data imbalance and limitations in data availability due to privacy concerns or storage constraints. By focusing on these realistic conditions in unlearning context, our algorithm not only fills a void in the current research but also sets a new benchmark for conducting class unlearning in these more complex scenarios. Our approach sheds some lights on the development of more robust, privacy-aware, and efficient unlearning techniques. The empirical results, as detailed in Section 5.1 of our paper, demonstrate the effectiveness of our method and its superiority over existing approaches in dealing with these challenges.
>
>
> We hope the above answer now comprehensively address the motivations behind our algorithm, both from practical and intellectual standpoints.
>
>
> **References**
>
> [1] Chundawat, Vikram S., et al. "Zero-shot machine unlearning." IEEE Transactions on Information Forensics and Security (2023).
>
> [2] Tarun, Ayush K., et al. "Fast yet effective machine unlearning." IEEE Transactions on Neural Networks and Learning Systems (2023).

---

> ### Author Response · Authors · 2023-11-16
> **Response to Reviewer iDvc (Part 3/6)**
>
> ### **Q3: Better Baselines**
>
> We appreciate your feedback regarding the incorporation of appropriate baselines in our study. We have conducted a review of the relevant literature and made every effort to include all necessary baselines that align with the scope of our research on class unlearning in imbalanced data scenarios. Our baseline selection reflects the most relevant and recent findings from top-tier conferences and journals over the past three years.
>
> Here is a summary of the baselines we have included, illustrating their ability in tackling the current unlearning problem. It is worth mentioning that none of them touch the challenging imbalanced data problem as we did.
>
> | Baseline        | No Retraining | Data Access Restriction |
> |-----------------|---------------|-------------------------|
> | GKT [1]         | Yes           | Yes                     |
> | UNSIR [2]       | Yes           | Yes                     |
> | Unrolling [3]   | Yes           | No                      |
> | I-R (Amnesiac) [4] | Yes       | No                      |
>
> While we believe our current selection of baselines provides a comprehensive comparison for our algorithm, we acknowledge the possibility of overlooking other significant works. Therefore, we kindly invite you to suggest any additional methods that you think are essential for comparison. We are committed to conducting a thorough and fair evaluation and are more than willing to include any recommended baselines in our analysis. Thank you in advance for your suggestion on a comprehensive evaluation of our proposed method.
>
> **References**
>
> [1] Chundawat, Vikram S., et al. "Zero-shot machine unlearning." IEEE Transactions on Information Forensics and Security (2023).
>
> [2] Tarun, Ayush K., et al. "Fast yet effective machine unlearning." IEEE Transactions on Neural Networks and Learning Systems (2023).
>
> [3] Thudi, Anvith, et al. "Unrolling sgd: Understanding factors influencing machine unlearning." Proceedings of the 7th IEEE European Symposium on Security and Privacy. 2022.
>
> [4] Graves, Laura, Vineel Nagisetty, and Vijay Ganesh. "Amnesiac machine learning." AAAI. 2021.

---

> ### Author Response · Authors · 2023-11-16
> **Response to Reviewer iDvc (Part 4/6)**
>
> ***In addition, we want to help clarify some of the concerns raised specifically in the Weaknesses session.***
>
> ### **W1: In the presented experiments, there is not a good way of knowing what constitutes a good delta in classification based on an unlearning request. In the results table, the authors show that after unlearning, the accuracy for the unlearned class is 0.0. I do not understand why there is any merit in this. Throughout the entire paper, there is never any mention of what constitutes a valid "forgetting" of a given class (as an exact definition). It seems that the implied definition (based on the results) is that the accuracy on the forgotten class should be zero. I disagree that this is a useful definition.**
>
> Thank you for your comment and the opportunity to clarify the concept and evaluation criteria of class unlearning as presented in our work.
>
> As stated in Section 1, Paragraph 2 of our paper, retraining is considered a gold standard for most unlearning problems in machine learning. The objective of class unlearning is to modify a trained model so that it behaves approximating or matching the performance of a model that has been retrained without the data from that class. One idea to assess this is to compare the accuracy of the unlearned model with that of the retrained model, whose accuracy on the forgetting class is zero. This standard is not unique to our work but is a well-established criterion in the field, as evidenced by studies in top-tier conference papers [1,2,3,4].
>
> In the context of our experiments, achieving an accuracy of 0.0 for the unlearned class is indicative of successful unlearning. This is because it demonstrates that the model no longer recognizes or correctly classifies instances of the unlearned class—essentially, the model has 'forgotten' this class. The closer the performance of the unlearned model on the unlearned class is to this zero accuracy, the more effective the unlearning process is considered.
>
> We followed this evaluation methodology, widely accepted and utilized in previous works, to ensure consistency and comparability of our results. In these studies, and in ours, the accuracy of the retrained model on the unlearned class typically approaches 0.0, indicating successful unlearning. Thus, our aim was to replicate this outcome in our unlearning algorithm to demonstrate its efficacy.
>
> We hope this explanation clarifies the rationale behind our evaluation. We believe this approach aligns with the current standards and practices in the field and provides a valid measure of the effectiveness of unlearning algorithms. We recognize, however, that the field of machine unlearning is rapidly evolving, and alternative perspectives on evaluation criteria may exist. We are open to considering other viewpoints and evaluation methods if suggested. Should you have specific recommendations or alternative criteria in mind, we would be more than willing to re-evaluate our methods and adapt our approach accordingly.
>
> **Reference**
>
> [1] Wang, Junxiao, et al. "Federated unlearning via class-discriminative pruning." WWW. 2022.
>
> [2] Graves, Laura, Vineel Nagisetty, and Vijay Ganesh. "Amnesiac machine learning." WWW. 2021.
>
> [3] Golatkar, Aditya, Alessandro Achille, and Stefano Soatto. "Eternal sunshine of the spotless net: Selective forgetting in deep networks." CVPR. 2020.
>
> [4] Bourtoule, Lucas, et al. "Machine unlearning." 2021 IEEE Symposium on Security and Privacy (SP). 2021.

---

> ### Author Response · Authors · 2023-11-16
> **Response to Reviewer iDvc (Part 5/6)**
>
> ### **W2.1  How imbalanced must the classes be for this method to work?**
>
> According to the question, we have conducted experiments to incorporate different imbalance rates (the rate that compares the number of minority class data to the number of majority class data) in the range of ${0.05, 0.1, 0.2, 0.4}$. Although we cannot show the results of all possible rates due to their continuous nature, the current results show a tendency for unlearning performance to decrease as the degree of imbalance increases in these ranges. The results are shown in the following table on the accuracy of retaining and forgetting data. We can see that even at a high imbalance rate, such as 0.05, our GENIU is still superior to other baselines.
>
> | r     | Acc                | origin | retrain | GKT    | UNSIR  | GENIU    |
> |-------|--------------------|--------|---------|--------|--------|--------|
> | 0.05   | $\mathcal{D}_r\uparrow$   | 0.7176 | 0.7573  | 0.3036   | 0.184 | 0.7227 |
> |        | $\mathcal{D}_f\downarrow$ | 0.9394 | 0.0     | 0.0   | 0.0   | 0.0653 |
> | 0.1   | $\mathcal{D}_r\uparrow$ | 0.8057 | 0.816   | 0.2595 | 0.3002 | 0.7711 |
> |       | $\mathcal{D}_f\downarrow$ | 0.9681 | 0.0     | 0.0    | 0.0016 | 0.0002 |
> | 0.2   | $\mathcal{D}_r\uparrow$ | 0.8591 | 0.8693  | 0.2593 | 0.7871 | 0.7509 |
> |       | $\mathcal{D}_f\downarrow$ | 0.954  | 0.0     | 0.0    | 0.0163 | 0.0    |
> | 0.4   | $\mathcal{D}_r\uparrow$ | 0.8908 | 0.9013  | 0.2241 | 0.8742 | 0.7706 |
> |       | $\mathcal{D}_f\downarrow$ | 0.9411 | 0.0     | 0.0    | 0.0031 | 0.0    |
>
> ### **W2.2 How large of a fraction of unlearning can this method support?**
>
> In the current experiemnts, the fraction of unlearning are varied in the range of [0.22, 0.53]. Even when half of the data is unlearned, our proposal still achieves superior performance. To the extreme, we try to forget the majoirty class, which comprises half of the original training data, and retain all other classes.

---

> ### Author Response · Authors · 2023-11-16
> **Response to Reviewer iDvc (Part 6/6)**
>
> ### **W4: The method seems overly complicated --- the authors fail to include simpler methods and show that they fail**
>
> Thank you for your feedback regarding the complexity of our proposed method. We understand the importance of a balance between technical depth and understandability. In our response, we aim to clarify our method, state why the complexity is both necessary and justified given the challenges we are addressing.
>
> In general, our method aligns with the pipeline of existing generative-based unlearning methods [1,2], utilizing generated proxies in the unlearning phase. The steps in our approach are as follows:
> 1. **Obtaining the Generator**: This occurs during the training phase, where actual data samples are available, allowing for balanced sampling and the generation of accurate proxies.
> 2. **Mini-Batch Tuning Strategy**: In the unlearning phase, we introduce a mini-batch tuning strategy to mitigate performance degradation, especially when forgetting the majority class in imbalanced settings.
> 3. **Optimizing the Tuning**: We use proxies close to the decision boundary of the original model to induce significant model updates.
>
> While our method incorporates these steps to address specific challenges, we also ensure to compare it with simpler baselines as shown in Section 5.2 and Appendix G. These compared baselines include this simpler versions of our proposal:
> 1. Training the generator only in the unlearning phase.
> 2. Implementing GENIU without the mini-batch tuning strategy.
> 3. Training the generator using min decision entropy samples.
>
> **Justification for Complexity:**
> The complexity of our method is a direct response to the intricate nature of the problem at hand. Simple methods, including those proposed in previous works or compared in the ablation study, have proven insufficient for effectively addressing the challenges of class unlearning in imbalanced data scenarios. Our experiments and the ablation study demonstrate the shortcomings of these simpler approaches and the potential need for a more complicated method like GENIU.
>
> **Response to 'Overly Complicated' Concern:**
> While we acknowledge that our method might initially appear complex, this complexity is not without purpose. It is tailored to address specific challenges inherent in the problem we are tackling. We will update the draft and try our best to present our method as clearly as possible, ensuring that its complexity does not hinder understandability. We believe that the term 'overly complicated' might not fully capture the deliberate and necessary design of our approach. However, we are open to suggestions on how to streamline or communicate the complexity of our methodology efficiently and welcome any specific feedback on this.
>
> We hope this explanation provides clarity on the rationale behind the design of our method and how it compares to simpler baselines. Our goal is to contribute a solution that is both robust and effective in addressing the unique challenges of class unlearning in imbalanced data scenarios.
>
> **References**
>
> [1] Chundawat, Vikram S., et al. "Zero-shot machine unlearning." IEEE Transactions on Information Forensics and Security (2023).
>
> [2] Tarun, Ayush K., et al. "Fast yet effective machine unlearning." IEEE Transactions on Neural Networks and Learning Systems (2023).

---

> ### Author Response · Authors · 2023-11-20
> **Looking forward to your further suggestions**
>
> Dear Reviewer iDvc, thank you for your time and thoughtful review. Following your suggestions, we have clarified our motivation and problem statement, and added experiments and analyses on the extent of data imbalance. We believe we have addressed your concerns in our response, and are open to further discussion if you have any remaining or further questions. We are keen to take the opportunity to answer any questions you may have before the discussion period closes. Thank you!

---

> ### Author Response · Authors · 2023-11-22
> **The latest revision has been uploaded.**
>
> Dear reviewers, I would like to thank you once again for your valuable review comments, which are very helpful in improving our work. We have carefully referred your comments and revised the paper accordingly, and the latest revised version has been uploaded. We would really appreciate the opportunity to receive your latest comments and discuss further before the discussion period closes. Much appreciate.

---

### Meta-Review · Area_Chair_CBFM · 2023-12-10

**Metareview:**

The paper proposes an innovative approach to addressing the challenge of class unlearning in scenarios with imbalanced data. However, after careful consideration of the reviewers' feedback and the authors' responses, the decision is to reject the paper in its current form due to several key issues.

Limited Novelty: Reviewer iDvc raised concerns about the novelty of the proposed method. Despite the innovative use of a VAE, the approach did not sufficiently distinguish itself from existing methods in the field, affecting the perceived contribution of the work.

Incomplete Experimental Details: The paper lacks essential experimental details, such as dataset sizes, validation sets, and runtime information. This information is crucial for assessing the empirical validation of the research, as pointed out by Reviewer iDvc.

Clarity Issues in Technical Details and Notation: Reviewer iMwM noted significant clarity issues in the technical presentation and notation. These issues create barriers to understanding the paper's methodology and potentially limit its reproducibility.

Concerns About Soundness and Contribution: Reviewer PTAU highlighted poor soundness and contribution, suggesting that the methodological foundations of the paper and its overall impact on the field are not adequately established.

Lack of Motivation: The reviewers expressed concerns about the lack of clear practical or intellectual motivation for the proposed algorithm. While the authors addressed this in their response, the initial submission did not effectively convey the significance and relevance of the research.

Baseline Comparisons: There were concerns about the selection of appropriate baselines for comparison. The reviewers suggested that the chosen baselines might not adequately reflect the current state-of-the-art, affecting the evaluation of the proposed method's performance.

**Justification For Why Not Higher Score:**

as stated in meta review

**Justification For Why Not Lower Score:**

NA

---

### Decision · Program_Chairs · 2024-01-16

Reject